# Sex differences in outcomes from mild traumatic brain injury eight years post-injury

**Nicola Jayne Starkey** [1]*, **Brittney Duffy**[1], **Kelly Jones**[2], **Alice Theadom**[2], **Suzanne Barker-Collo**[3], **Valery Feigin**[2], **on behalf of the BIONIC8 Research Group**¶

**1** School of Psychology, University of Waikato, Hamilton, New Zealand, **2** National Institute for Stroke and Applied Neurosciences, School of Clinical Sciences, Auckland University of Technology, Auckland, New Zealand, **3** School of Psychology, University of Auckland, Auckland, New Zealand

☯ These authors contributed equally to this work.
¶ Membership of the BIONIC8 Research Group is provided in the Acknowledgments.
* nicola.starkey@waikato.ac.nz

**Data Availability Statement:** There are ethical restrictions on sharing of de-identified data for this study. We do not have the participants' or ethics committee's permission to share their anonymous

## Abstract

The long-term effects of mild TBI (mTBI) are not well understood, and there is an ongoing debate about whether there are sex differences in outcomes following mTBI. This study examined i) symptom burden and functional outcomes at 8-years post-injury in males and females following mTBI; ii) sex differences in outcomes at 8-years post-injury for those aged <45 years and ≥45 years and; iii) sex differences in outcomes for single and repetitive TBI. Adults (≥16 years at injury) identified as part of a population-based TBI incidence study (BIONIC) who experienced mTBI 8-years ago (N = 151) and a TBI-free sample (N = 151) completed self-report measures of symptoms and symptom burden (Rivermead Post-Concussion Symptom Questionnaire, Hospital Anxiety and Depression Scale, Post-traumatic Stress Disorder Checklist), and functional outcomes (Participation Assessments with Recombined Tools, Work Limitations Questionnaire). The mTBI group reported significantly greater post-concussion symptoms compared to the TBI-free group (F(1,298) = 26.84, p<.01, ηp² = .08). Females with mTBI were twice as likely to exceed clinical cut-offs for post-concussive (X²(1)>5.2, p<.05, V>.19) and PTSD symptoms (X²(1) = 6.10, p = .014, V = .20) compared to the other groups, and reported their health had the greatest impact on time-related work demands (F(1,171) = 4.36, p = .04, ηp² = .03. There was no interaction between sex and age on outcomes. The repetitive mTBI group reported significantly greater post-concussion symptoms (F(1,147) = 9.80, p<.01, ηp² = .06) compared to the single mTBI group. Twice the proportion of women with repetitive mTBI exceeded the clinical cut-offs for post-concussive (X²(1)>6.90, p<.01, V>.30), anxiety (X²(1)>3.95, p<.05, V>.23) and PTSD symptoms (X2(1)>5.11, p<.02, V>.26) compared with males with repetitive TBI or women with single TBI. Thus, at 8-years post-mTBI, people continued to report a high symptom burden. Women with mTBI, particularly those with a history of repetitive mTBI, had the greatest symptom burden and were most likely to have symptoms of clinical significance. When treating mTBI it is important to assess TBI history, particularly in women. This may help identify those at greatest risk of poor long-term outcomes to direct early treatment and intervention.

data. The dataset includes sensitive information, and given the nature of the data and the relatively small region of data collection the participants may be identifiable despite efforts to annonymise the data. Qualified and interested researchers may request access to the data by contacting University of Waikato Ethics Committee (humanethics@waikato.ac.nz.)

**Funding:** NJS, BD, KJ, AT, SBC, VF WMRF #289. Waikato Medical Research Foundation. https:// wmrf.org.nz/ VF, AT, SBC, NJS 09/063A. Health Research Council of New Zealand. https://www. hrc.govt.nz/grants-funding. The funders had no role in study design, data collection and analysis, decision to publish, or preparation of the manuscript.

**Competing interests:** The authors have declared that no competing interests exist.

## Introduction

It is estimated that more than 10 million people experience a TBI each year, with 70–95% of these being classified as mild [1, 2]. TBI has widespread impacts [3], and whilst the effects of moderate and severe TBI are well documented, much less is known about the long-term outcomes of mild TBI (mTBI). This may be because a significant proportion do not seek medical treatment at the time of injury and the term 'mild' implies little or no impact [2, 4]. Rehabilitation and recovery continue for many years post-TBI [5, 6], and the economic, personal, family and societal costs of sustaining even an mTBI can be substantial [7–9].

Symptoms of TBI vary but typically include what are commonly known as post-concussive symptoms. These are grouped into cognitive (e.g., forgetfulness, poor concentration, difficulty thinking), somatic (e.g., nausea, fatigue, dizziness), and emotional symptoms (e.g., irritability, feeling depressed, feeling frustrated). The majority of people who sustain an mTBI recover within three months of injury however, there is evidence that 46–65% of patients report persistent symptoms at 6 months [10, 11], and 10–53% of patients continue to experience persistent symptoms up to and beyond 1-year post-injury [12–15]. At four years post-injury, people with mTBI in the BIONIC cohort reported significantly more cognitive symptoms than community controls [16]. Neuropsychiatric problems including depression, anxiety, and post-traumatic stress disorder (PTSD) are also reported following mTBI, with anxiety and depression diagnosed more commonly than PTSD [17, 18]. In a longitudinal study of outcomes from mTBI, 38.1% of the sample met the cut-offs for anxiety and or depression on the HADS at 12 months post-injury (2.2% depression, 25.5% anxiety, 10.2% depression and anxiety) which decreased to 32.1% four years post-injury (7.7% depression only, 13.6% anxiety only and 11.8% depression and anxiety), similar to rates observed in community samples [19].

MTBI can also negatively impact functional outcomes including work and community engagement. The CENTER-TBI study reported that the majority of participants with few post-concussive symptoms had returned to work 6 months post-injury (83.2%). However, those with higher concussive or PTSD symptom burden had much lower return to work rates (46.1% - 62.5%), and some (17.2%) were working reduced hours [20]. The work-related impacts of mTBI appear to be quite persistent; four-year post-injury outcomes from the BIONIC cohort found that people with mTBI experienced more difficulty at work because of their health compared to the New Zealand (NZ) general population [21]. Also, 17.3% of people with mTBI left the workforce or reduced their hours within 4-years of injury, and 15.5% reported that their work was negatively affected by the injury [21]. Similar findings are reported from the Danish national concussion registry. Those with concussion were less likely to be working five years post-injury than matched controls (43% vs 30%) [22]. Participation in other everyday activities (e.g. socializing with friends, carrying out community and home-based activities) is also negatively impacted following mTBI [16].

These findings highlight the potential negative long-term impacts of mTBI and suggest the need for further longer-term follow-up studies to evaluate the full effect of mTBI across a range of outcomes and the need for long-term monitoring to provide appropriate support and rehabilitation when required.

Whilst a range of factors, including age, minority ethnicity, lower levels of education, lower socioeconomic status (SES), and previous TBI are linked to poorer outcomes [16, 23–27], one of those which is least understood is sex. The findings regarding sex differences and TBI are inconsistent, and, the majority of studies have been conducted with specific groups (e.g., athletes, military) rather than with a general population sample. A systematic review reported that women experience a greater symptom burden 12- to 18-months post-injury than men [28], and the proportion of women experiencing persistent symptoms was greater than would be

expected based on incidence [29]. A recent review concluded that women endorse more con-cussion symptoms than men, and evidence regarding sex differences in recovery rates or in relation to impairments in specific domains was inconclusive [30]. The authors highlighted the need for further work beyond the sports domain, with appropriate comparison groups and a longer follow-up period.

Several recent studies have explored sex differences in general population mTBI samples. The CENTER-TBI team found that 6-months after mTBI, women were at higher risk of poorer functional outcomes, were less likely to return to work, had greater post-concussion, depres-sion and anxiety symptom burden and poorer health-related quality of life compared to men [31]. Outcomes varied by age, with women aged <45 and >65 years having the poorest out-comes compared to men [31]. The TRACK-TBI team also recently reported an interaction between sex and age with PTSD and post-concussion symptoms 6-months after mTBI in young adults (18–39 years) [32]. Females in the oldest age group (30–39 years) reported more symptoms compared to the other groups (males 18–29 or 30–39 years, females 18–29 years) [32]. At 12 months post-injury, women reported significantly more severe cognitive and somatic symptoms, a pattern not observed in an orthopaedic trauma control group. In addi-tion, women with mTBI aged 35–49 years had more severe somatic symptoms compared with younger (17–34 years) or older participants (<50 years) [33]. A recently published 10 year fol-low up of health-related quality of life (HRQoL) after mild, moderate, or severe TBI found that a significantly smaller proportion of females (51.5%) reported good HRQoL compared to males (68.6%). Outcomes were worst in females aged 54–76 years, suggesting that older females may be particularly vulnerable to poor long-term outcomes [34]. Whilst these studies provide interesting insights into sex differences in outcomes from mTBI, it is still unclear which age group is likely to have the poorest outcomes. In addition, previous studies have sev-eral limitations, including the lack of a control group, a limited follow-up period and assess-ment of a limited range of outcomes [4]. To address these limitations, we report the 8-year outcomes following mTBI in a population-based cohort of adults (≥16 years at injury) com-pared to an age- and sex-matched TBI free sample [2]. The primary aim was to compare symp-toms and symptom burden and functional outcomes in males and females with mTBI with an age- and sex-matched TBI-free comparison group. Secondary aims were to 1) compare out-comes at 8 years post-injury in males and females aged <45 years and ≥45 years (at injury) and 2) compare outcomes in those with single and repetitive TBI.

## Method

### Participants and study design

The study received approval from NZ Health and Disability Ethics Committees (#17/STH/247).

Participants with mTBI were identified through the BIONIC study, an epidemiological population-based incidence study of all cases of TBI in Hamilton, New Zealand, between 1 March 2010 and 28 February 2011 [2]. TBI was defined according to the World Health Orga-nisation (WHO) criteria as "an acute brain injury resulting from mechanical energy to the head from external physical forces" [35]. Operational criteria for identification of TBI included the presence of one or more of: confusion or disorientation; loss of consciousness; post-trau-matic amnesia; other neurological abnormalities (e.g. focal neurological signs, lesions or sei-zure) not due to drugs, alcohol, medication or caused by other injuries [2, 35]. MTBI was defined as a Glasgow Coma Scale (GCS) score of 13–15 and/or post-traumatic amnesia (PTA) of less than 24 hours [36]. Each mTBI case was further classified as having low, medium or high risk of developing intracranial hematoma based on the presence or absence of clinical

findings (loss of consciousness, amnesia, vomiting or diffuse headache), neurological deficits, (impaired vision, hearing, speech, balance, walking difficulties or weakness), skull fracture and other risk factors (coagulopathy, drug/alcohol consumption, previous neurosurgical procedures, pre-trauma epilepsy, over 60 years of age) [37]. MTBI cases with no clinical or neurological deficits, no skull fracture or risk factors were classified as low risk. Those with clinical findings but no neurological deficits, skull fracture or risk factors were classed as medium risk. The high-risk group included those with clinical, neurological, skull fracture and/or other risk factors.

TBI cases were identified using multiple methods, including checks of hospital admissions, general practitioners, sports centres, community health centres, and self-referrals. The national accident compensation provider (ACC) databases were also screened to ensure that all cases of TBI (particularly mild injuries) were identified (see [38] for further details). For cases identified through the hospital or a medical provider, demographic and injury details were extracted from medical records and verified when participants were invited to take part in the TBI outcome study.

This study focused on those with mTBI aged ≥16 years at injury (i.e., ≥ 24 years of age at assessment). As shown in Fig 1, 341 participants completed assessments at 6/12 months post-injury, 346 were approached for the 8 year follow up (participants who had missed earlier assessments were also contacted), and 151 (43.6%) completed the 8-year post-injury assessment. The characteristics of the incidence sample and those completing the 8-year outcome assessments are shown in Table 1.

An age- and sex-matched cohort (n = 151), free from TBI in their lifetime, were recruited for comparison purposes (Table 2). Participants were recruited from a panel maintained by ResearchNow, an online data collection company. Participants in the comparison group were screened for previous TBI or injuries likely to result in a TBI (e.g., car accident, assault resulting in hospitalisation) prior to completing the questionnaire.

## Measures

Participants completed a set of study specific questionnaires to gather demographic information, a screening measure to assist with recall of TBI, and a set of patient reported outcome measures (PROMS) assessing symptoms and symptom burden (post-concussion, depression, anxiety and PTSD symptoms) and functional status (community participation and work related productivity) [39].

**Symptoms and symptom burden.** The Rivermead post-concussive symptoms questionnaire (RPQ) is a 16-item self-report questionnaire designed to assess common symptoms after a head injury [40]. Participants rate how much of a problem each symptom is from 0 = not experienced, 2 = a mild problem, 3 = a moderate problem, to 4 = a severe problem. Participants rated their symptoms over the last 24 hours (as in [41]) rather than comparing their symptoms to before their injury. This was to improve the accuracy of the answers for those with mTBI (it was 8 years since their injury) and to make the questions applicable for the comparison group. (Note: the response of 1 = no more of a problem was excluded because it did not make sense as the participants were not comparing how they felt now to an earlier time point and scores of 1 are excluded from the calculation of the total score). The total RPQ score was calculated, and subscale scores for the cognitive, emotional and somatic domains. A score of 3 or more on four items was used to determine if participants met the DSM-IV symptom criteria for post-concussive disorder [42].

The Hospital Anxiety and Depression Scale (HADS) is a 14- item self-report questionnaire designed to measure levels of anxiety and depression [43]. It has high internal consistency, and

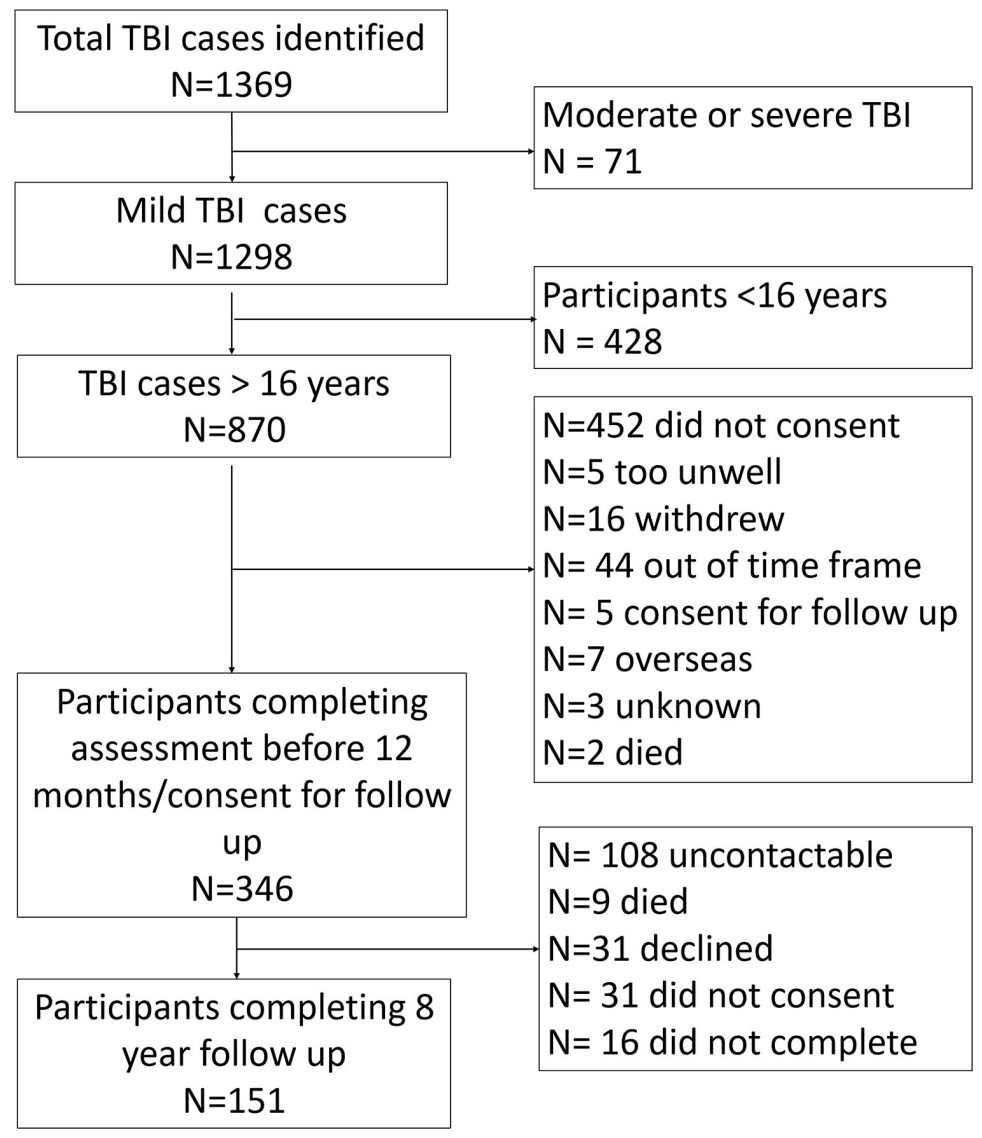

**Fig 1. Participant flow through the study.**

sensitivity to change and has been extensively used with TBI populations [44, 45]. The HADS has two subscales, Anxiety and Depression, containing seven items each. For each item, respondents are asked to indicate the extent to which each statement applied to them over the past week. Responses are scored on a 4-point Likert scale (0 to 3) for each question. The minimum score for each subscale is 0, and the maximum score is 21, with higher scores indicating greater symptom severity. Scores of 8 and above are suggestive of clinical levels of depressive or anxiety symptoms.

The abbreviated Post-traumatic stress disorder checklist: Civilian scale (PCL-C) was used to screen for the presence and severity of post-traumatic stress symptoms. The measure was developed for use in primary care and has been validated for men and women aged 18–80 years [46]. To reduce participant burden, the abbreviated 6-item version was used (sensitivity = .95, specificity = .69 with a cut-off of 14) [47]. Participants indicate how much they have

**Table 1. A comparison of the demographic and injury characteristics of the Bionic incidence study cohort and the sample consenting to the 8-year follow-up.**
Mean (standard deviation) or number (percentage) of participants for demographic and injury variables for the TBI incidence cohort and the sample consenting to the eight year follow up. Chi-square goodness of fit tests were used to determine if the 8-year sample was representative of the incidence sample.

| | | Incidence sample (n = 870) | 8-year sample (n = 151) | Comparison |
|---|---|---|---|---|
| **Sex** | Male, n (%) | 533 (61.3) | 77 (51.0) | $X^2(1) = 6.71$, p = .01[*] |
| **Age** | Mean (SD) | 37.37 (19.28) | 37.58 (16.99) | t(150) = 0.15, p = .88 |
| **Ethnicity** | European | 543 (62.4) | 115 (75.5) | $X^2(2) = 12.58$, p<.01[*] |
| | Māori | 254 (29.2) | 30 (19.9) | |
| | Other | 73 (8.4) | 6 (4.6) | |
| **MTBI severity** | Low risk | 155 (17.8) | 19 (12.6) | $X^2(2) = 2.84$, p = .24 |
| | Medium risk | 193 (22.2) | 36 (23.8) | |
| | High risk | 522 (60.0) | 96 (63.6) | |
| **Prior TBI** | None | 266 (30.6) | 61 (40.1) | $X^2(2) = 2.96$, p = .23 |
| | 1 or 2 | 179 (20.5) | 56 (37.1) | |
| | 3 or more | 105 (12.1) | 30 (19.9) | |
| | Unknown | 320 (36.8) | 4 (2.6) | |
| **Injury mechanism** | Fall | 274 (31.5) | 49 (32.5) | $X^2(4) = 1.55$, p = .82 |
| | Traffic/MVA | 192 (22.1) | 39 (25.8) | |
| | Mech Force | 178 (20.5) | 30 (19.9) | |
| | Assault | 190 (21.8) | 30 (19.9) | |
| | Other | 12 (1.5) | 3 (2) | |
| | Unknown[a] | 24 (2.8) | 0 | |
| **CT scan conducted** | | 142 (16.3) | 24 (15.9) | |
| **GCS** | 13 | 16 (1.8) | 5 (3.3) | $X^2(3) = 68.15$, p<.01[*] |
| | 14 | 14 (1.3) | 16 (10.6) | |
| | 15 | 398 (45.7) | 67 (44.4) | |
| | Unknown[a] | 353 (40.6) | 63 (41.7) | |
| **Education** | Primary School | 19 (2.2) | 2 (1.3) | $X^2(3) = 12.77$, p <.01[*] |
| | High School | 198 (22.8) | 56 (37.1) | |
| | Polytechnic | 108 (12.4) | 41 (27.2) | |
| | University | 90 (10.3) | 46 (30.5) | |
| | Unknown[a] | 455 (52.3) | 6 (4.0) | |
| **Occupation** | Professional | 69 (7.9) | 34 (22.5) | $X^2(2) = 6.61$, p = .04[*] |
| | Skilled | 119 (13.7) | 44 (29.1) | |
| | Other | 225 (25.9) | 65 (43.0) | |
| | Unknown[a] | 413 (47.5) | 8 (5.3) | |

[a] 'unknown' cases were excluded from the chi square analysis.

[*] p<.05.

[**] <.01.

GCS = Glasgow Coma Scale score. CT = computerised tomography

been bothered by six problems (e.g., disturbing memories from a past event) from 1 (not at all) to 5 (extremely). A positive screen is a score of ≥14.

**Functional status.** The Participation assessment with recombined tools–objective (PART-O) was used to assess participation in the community and everyday life. The original Part-O was developed to integrate three commonly used measures for people with TBI (Community Integration Questionnaire [48], Participation Objective, Participation Subjective [49], and the Craig Assessment and Reporting Technique [50]). The revised 17 item self-report tool measures three domains of community participation (Productivity, Out and About, and Social

**Table 2. A comparison of the demographic characteristics of the males and females in the mTBI group 8 years post-injury and the comparison group.** Data are presented as mean (standard deviation) or number (percentage) for demographic variables for the mTBI and comparison group and for injury severity and medical diagnoses for the mTBI group (information not available for the comparison group). Two-way ANOVA or chi-square tests were used to explore differences by group and gender.

| | | mTBI | | | Comparison | | | Statistical test |
|---|---|---|---|---|---|---|---|---|
| | | Male (n-77) | Female (n = 74) | Total (n = 151) | Male (n = 74) | Female (n = 77) | Total (n = 151) | |
| **Age** (mean, SD) | | 43.23 (16.15) | 47.69 (17.78) | 45.42 (17.05) | 46.84 (15.13) | 47.10 (16.34) | 46.97 (15.75) | Group $F_{(1,298)}$ = 0.64, p = .42, $\eta p^2$<.01 Gender F (1,298) = 1.57, p = .21, $\eta p^2$<.01 Int (F(1, 298) = 1.23, p = .27, $\eta p^2$<.01 |
| **Ethnicity** (n, %) | NZ European | 62 (80.5) | 53 (71.6) | 115 (76.2) | 62 (83.8) | 72 (93.5) | 134 (88.7) | $X^2$ = (1) = 1.45, p = .23, V = .08 |
| | Māori | 14 (18.2) | 16 (21.62) | 30 (19.9) | 7 (9.5) | 2 (2.6) | 9 (6.0) | $X^2$ = (1) = 2.70, p = .14, V = .26[a] |
| | Other | 1 (1.3) | 5 (6.8) | 6 (4.0) | 4 (5.4) | 3 (3.9) | 7 (4.6) | $X^2$ = (1) = 2.24, p = .26, V = .42[a] |
| **Education** (n, %) | Degree | 18 (23.3) | 22 (29.7) | 40 (26.5) | 27 (36.5) | 23 (29.9) | 50 (33.1) | $X^2$ = (1) = 0.72, p = .40, V = .09 |
| | Diploma | 24 (31.2) | 30 (40.5) | 54 (35.8) | 15 (20.3) | 18 (23.4) | 33 (21.9) | $X^2$ = (1) = 0.01, p = .93, V = .01 |
| | Trade/Tech | 24 (31.2) | 3 (4.1) | 27 (17.9) | 21 (28.4) | 12 (15.6) | 33 (21.9) | **$X^2$ = (1) = 5.05, p = .03, V = .29**[*] |
| | Other | 11 (14.3) | 19 (25.7) | 30 (19.9) | 11 (14.9) | 24 (31.2) | 35 (23.2) | $X^2$ = (1) = 0.20, p = .66, V = .06 |
| **Occupation** (n, %) | Professional | 27 (35.1) | 20 (27.0) | 47 (31.2) | 27 (26.5) | 21 (27.3) | 48 (31.8) | $X^2$ = (1) = 0.01, p = .91, V = .01 |
| | Skilled | 18 (23.4) | 12 (16.2) | 30 (19.9) | 18 (24.3) | 17 (22.1) | 35 (23.2) | $X^2$ = (1) = 0.48, p = .49, V = .09 |
| | Other | 31 (40.3) | 38 (51.4) | 69 (45.7) | 29 (39.2) | 39 (50.6) | 68 (45.03) | $X^2$ = (1) = 0.07, p = .79, V = .02 |
| **Main earner** (n, %) | Yes | 40 (51.9) | 24 (32.4) | 64 (42.4) | 36 (48.6) | 22 (28.6) | 58 (38.4) | $X^2$ = (1) = 0.02, p = .96, V = .01 |
| | Missing | 17 (22.1) | 27 (36.5) | 44 (29.1) | 25 (33.8) | 28 (36.4) | 53 (35.1) | |
| **mTBI severity** (n, %) | Low risk | 8 (10.4) | 11 (14.9) | 19 (12.6) | | | | $X^2$ = (2) = .90, p = .64, V = .08 |
| | Medium risk | 20 (26.0) | 16 (21.6) | 36 (23.8) | | | | |
| | High risk | 49 (63.6) | 47 (63.5) | 96 (63.6) | | | | |
| **Other health diagnoses** (n, %) | Cardiovascular | 9 (11.6) | 9 (12.2) | 18 (11.9) | - | - | - | $X^2$ = (1)) = 0.01, p = .56, V < .01 |
| | Dementia | 0 | 1 (1.4) | 1 (0.7) | - | - | - | $X^2$ = (1) = 1.05, p = .49, V = .08[a] |
| | ADD | 0 | 1 (1.4) | 1 (0.7) | - | - | - | $X^2$ = (1) = 1.05, p = .49, V = .08[a] |
| | Neurological | 2 (2.6) | 2 (2.7) | 4 (2.7) | - | - | - | $X^2$ = (1) = 0.01, p = 1.00, V < .01 [a] |
| | Psychiatric | 6 (7.8) | 11 (14.9) | 17 (11.3) | - | - | - | $X^2$ = (1) = 1.89, p = .20, V = .11 |
| | Diabetes | 1 (1.3) | 3 (4.1) | 4 (2.7) | - | - | - | $X^2$ = (1) = 1.11, p = .36, V = .09 [a] |
| | Seizure | 0 | 4 (5.4) | 4 (2.7) | - | - | - | $X^2$ = (1) = 4.28, p = .06, V = .17 [a] |
| | Chronic fatigue | 1 (1.3) | 1 (1.4) | 2 (1.3) | - | - | - | $X^2$ = (1) = 0.01, p = 1.00, V < .01 [a] |
| | Other | 19 (24.7) | 20 (27.0) | 39 (25.8) | - | - | - | $X^2$ = (1) = 0.11, p = .85, V = .03 |
| | Missing | 2 (2.6) | 2 (2.7) | 4 (2.7) | - | - | - | |
| | None reported | 37 (48.1) | 20 (27.0) | 57 (37.7) | - | - | - | |

[a] = Fischer's Exact Test as some cell sizes <5.

[*]p<.05.

$\eta p^2$ = partial eta squared. V = Cramer's V.

Relations) designed specifically for use in populations with TBI [51]. The items ask about the length of time or frequency of engagement in various activities in a typical week (e.g., cleaning, cooking). An average score is calculated for each domain.

The Work Limitations Questionnaire (WLQ) was developed to measure the impact of chronic health problems and/or treatment on work-related productivity [52]. The self-report questionnaire was validated in populations working at least 20 hours a week with various chronic health conditions, including respiratory, gastrointestinal and psychiatric disorders. The 25-item questionnaire has four subscales: Time Demands (e.g., work required hours, 5 items), Physical Demands (e.g., perform physical job-related tasks, 6 items), Mental/Interpersonal Demands (e.g., concentrate on your work, 9 items) and Output Demands (e.g., being

able to cope with workload, 5 items). Participants respond by choosing one of five options from 0% (health makes the job demand difficult none of the time) to 100% (health makes the job demand difficult all of the time), or the item does not apply to their job. The subscales scores are the mean of the responses for the items within the scale, and indicate the percentage of the time their health makes their job difficult. The scale has good internal reliability and validity and has been used previously in TBI populations [21].

A screening questionnaire supplemented the PROMS to identify TBI in the comparison sample (the Ohio State University TBI Identification Method; OSU TBI-ID) [53]. Participants were asked five probe questions (e.g., '*have you ever injured your head or neck in a fall or from being hit by something, or by playing sports or on the playground*?'). Two additional questions followed each positive response: '*did you lose consciousness*?' and '*did you feel dazed or have a gap in your memory*?' Participants in the comparison group indicating loss of consciousness (LOC) or feeling dazed or confused after any incident were ineligible for the study. All participants were also asked for demographic information, and information about their living and work situations; those in the mTBI group were also asked about other health or medical conditions. All questionnaire responses were captured electronically via the Qualtrics platform.

## Procedure

Participants with mTBI who had agreed to follow-up were contacted by phone or email 8 years (plus or minus two months) after their injury. Updated contact details were sought from next of kin, health care providers, and the electoral roll when necessary. Participants were given information about the study over the phone, and a copy of the participant information sheet and consent form was emailed or posted to them. Participants completed the measures online or face-to-face. Participants in the mTBI group received a $10 supermarket voucher as a thank you.

Participants in the comparison group were recruited through ResearchNow. Those interested in completing the survey were screened for previous head injuries using the OSU-TBI questionnaire. Eligible participants completed the measures online and were recompensed at a standard rate by the market research company.

## Statistical analysis

Data were imported from Qualtrics into IBM SPSS version 27 and screened for missing values and outliers. Chi-square goodness of fit tests were conducted to check the representativeness of the sample participating in the study at 8 years with the original incidence sample. Descriptive statistics, supplemented by chi-square or ANOVA, were used to compare the demographic characteristics of the males and females in the mTBI and comparison group. Data from the RPQ, HADS and PCL-C were positively skewed. However, as the data were similarly skewed for each group, the sample sizes were reasonable, the data did not violate the homogeneity of variance assumptions, and ANOVA is considered robust to violations of normality; we considered it appropriate to use parametric analyses [54, 55].

A series of 2 (Group: mTBI, Comparison) x 2 (Sex: Male, Female) x 2 (Age: 16–44 years, ≥45 years at injury) between groups ANOVAs were conducted to examine differences between the groups for the PROMS. The age groups were selected because previous studies reported poorer outcomes in those aged <45 years [31], and this resulted in two relatively evenly sized groups for the analysis. Where the initial analyses showed no significant Sex x Age group or Group x Sex x Age group interactions, the ANOVAs were re-run without age to simplify interpretation and reporting (i.e., as 2 x 2 ANOVAs). When ANOVA showed a significant main effect of Sex or a Sex x Group interaction, additional ANOVAs were conducted for each item on the measure to identify the symptoms that differed in men and women with

mTBI (the Bonferroni correction was used to adjust for multiple comparisons). For brevity, only statistically significant findings are reported in the main text for the item analysis (see S1 Table for the full analyses). For the ANOVA, partial eta squared ($\eta p^2$) of 0.01 was interpreted as a small effect, 0.06 indicated a medium effect and 0.14 a large effect [56]. Further analyses (chi-square tests) were conducted to examine the proportion of males and females exceeding clinical cut-offs for the RPQ, HADS and PCL-C. Cramer's V of .01 indicates a small effect, .30 a medium effect and .50 a large effect [56].

Finally, to explore if repetitive TBI was associated with poorer outcomes, a series of 2 (Group: repetitive TBI, single TBI) x 2 (Sex: Male, Female) ANOVAs were conducted. Chi-square tests were conducted to determine if the percentage of males and females exceeding the clinical cut-offs for the RPQ, HADS, and PCL-C in the single and repetitive TBI groups differed.

## Results

The demographic and injury details of the original incidence sample and those assessed at 8 years post-injury are presented in Table 1. The sample completing the 8-year outcome assessments was comparable to the incidence cohort for age, mTBI severity, number of prior TBI and injury mechanism. However, the 8-year sample had a significantly greater proportion of females, those of NZ European ethnicity, participants with a GCS of 14, and those with higher levels of education, and professional or skilled jobs. [Note: there was a large amount of missing data for the incidence sample for information not contained in the medical records (prior TBI, education and occupation), and GCS was not recorded in many cases].

The demographic characteristics of the males and females in the mTBI and the comparison group are shown in Table 2. There were no significant differences in age, or associations between the groups for age, ethnicity, levels of education (apart from trade/technical certificate), occupation, or being the primary income earner. However, there were medium effect sizes for the chi-square tests comparing the proportion of participants of Māori and Other ethnicities. This may be due to the higher proportion of Māori participants in the mTBI group and only a single male participant of 'Other' ethnicity in the mTBI group. A significantly greater proportion of men held a trade or technical certificate compared to women for both the mTBI and comparison group (medium effect size). There was no association between sex and injury severity or with pre-existing or comorbid medical conditions in the mTBI group, although the effect sizes for psychiatric and seizure diagnoses were in the small to medium range, with greater numbers of women being diagnosed with these conditions.

The 3-way ANOVAs, including age as a factor, did not show significant Sex x Age interactions; therefore the results of the 2-way ANOVAs (Group x Sex) are reported (Table 3).

### Symptoms and symptom burden

The mTBI group obtained significantly higher scores with medium to large effect sizes for the RPQ total cognitive, and emotional subscales compared to the TBI-free group, indicative of greater post-concussion symptom burden (Table 3). Women obtained significantly higher RPQ total and somatic subscales scores than men (medium effect size). Fig 2a shows the proportion of males and females exceeding the cut-off scores for the RPQ, HADS and PCL-C. For the RPQ, a significantly greater proportion (small to medium effect) of the mTBI participants met the DSM-IV symptom criteria for post-concussive disorder (30.5%, n = 46) than the comparison group (13.2%, n = 20; $X^2$ (1) = -13.11, $p <$.01, V = .21). A greater proportion of men in the mTBI group were above the DSM-IV symptom cut-off compared to the TBI-free group, but this difference was not statistically significant with a small effect size ($X^2$ (1) = 3.47, $p$ = .06, V = .15). A significantly greater proportion of women (medium effect size) in the mTBI group

**Table 3. Descriptive and inferential statistics for the males and females in the mTBI (8-years post-injury) and comparison groups for the outcome measures.** Data are presented as mean (standard deviation) for males and females in the mTBI group 8 years post-injury and a TBI-free comparison group for measures of symptom and symptom burden and functional outcomes. Two-way ANOVAs were conducted to explore differences by group and gender.

| | mTBI at 8 years M (SD) n = 151 | | Comparison M (SD) n = 151 | | Two way ANOVA | | |
|---|---|---|---|---|---|---|---|
| | Males (n = 77) | Females (n = 74) | Males (n = 74) | Females (n = 77) | Group (F, sig, partial eta square) | Sex (F, sig, partial eta square) | Group x Sex (F, sig, partial eta square) |
| **Post-Concussion Symptoms (RPQ)** (df = 1, 298) | | | | | | | |
| Total | 14.00 (12.92) | 20.54 (16.49) | 7.97 (10.28) | 11.15 (11.18) | **26.84, p<.01, $\eta p^2$ = .08**\*\* | **10.68, p<.01, $\eta p^2$ = .04**\*\* | 1.27, p = .26, $\eta p^2$<.01 |
| Cognitive | 3.53 (3.52) | 4.53 (3.90) | 1.58 (2.90) | 1.96 (2.972) | **35.56, p<.01, $\eta p^2$ = .11**\*\* | 3.29, p = .07, $\eta p^2$ = .01 | 0.66, p = .42, $\eta p^2$<.01 |
| Somatic | 4.53 (4.80) | 7.76 (6.83) | 3.99 (5.17) | 6.48 (6.65) | 1.80, p = .318, $\eta p^2$<.01 | **17.71, p<.01, $\eta p^2$ = .06**\*\* | 0.29, p = .59, $\eta p^2$<.01 |
| Emotional | 3.83 (4.13) | 4.93 (4.41) | 2.741 (3.50) | 2.71 (3.12) | **17.18, p<.01, $\eta p^2$ = .06**\*\* | 2.857, p = .11, $\eta p^2$<.01 | 0.81, p = .3, $\eta p^2$<.01 |
| **Mood (HADS and PCL)** (df = 1, 298) | | | | | | | |
| Depression | 4.00 (3.90) | 4.68 (4.15) | 3.50 (3.88) | 3.29 (3.34 | **4.31, p = .04, $\eta p^2$ = .01**\* | 0.32, p = .57, $\eta p^2$<.01 | 1.11, p = .29, $\eta p^2$<.01 |
| Anxiety | 4.58 (3.67) | 6.04 (4.40) | 4.84 (3.983) | 5.743 (4.02) | 0.15, p = .70, $\eta p^2$<.01 | **4.97, p = .03, $\eta p^2$ = .02**\* | 0.90, p = .35 $\eta p^2$<.01 |
| PTSD | 9.66 (4.01) | 11.78 (5.75) | 9.64 (4.80) | 9.06 (3.44) | **4.44, p = .04, $\eta p^2$ = .02**\* | **3.93, p = .04, $\eta p^2$ = .01**\* | **4.22, p = .04, $\eta p^2$ = .01**\* |
| **Community Participation (Part-O)** (df = 1, 293)[e] | | | | | | | |
| Productivity | 2.12 (0.94) | 2.16 (1.16) | 1.97 (1.01) | 2.05 (0.98) | 1.09, p = .87, $\eta p^2$<.01 | 0.19, p = 665, $\eta p^2$<.01 | 0.03, p = .78 $\eta p^2$<.01 |
| Social Relations | 2.52 (1.01) | 2.60 (0.89) | 2.28 (0.94) | 2.37 (0.83) | **5.02, p = .03, $\eta p^2$ = .02**\* | 0.39, p = .54, $\eta p^2$<.01 | 0.03, p = .87, $\eta p^2$<.01 |
| Out and About | 1.32 (0.63) | 1.24 (0.65) | 1.25 (0.55) | 1.24 (0.66) | 0.25, p = .62, $\eta p^2$<.01 | 0.42, p = .52, $\eta p^2$<.01 | 0.25, p = .62, $\eta p^2$<.01 |
| **Work Limitations (WLQ)** (df = 1, 171) | | | | | | | |
| Time Demands | 8.87[a] (10.14) | 18.31[b] (20.98) | 12.52[c] (13.60) | 11.81[d] (18.74) | 0.35, p = .56, $\eta p^2$<.01 | 3.22, p = .08, $\eta p^2$ = .02 | **4.36, p = .04, $\eta p^2$ = .03**\* |
| Physical Demands | 11.79 (22.07) | 18.90 (25.65) | 9.83 (15.92) | 8.63 (15.97) | **4.00, p<.05, $\eta p^2$ = .02**\* | 1.85, p = .18, $\eta p^2$ = .01 | 1.85, p = .18, $\eta p^2$ = .01 |
| Mental/Interpersonal Demands | 9.61 (11.02) | 15.90 (16.57) | 9.95 (12.90) | 11.08 (18.22) | 1.01, p = .32, $\eta p^2$<.01 | 2.92, p = .09 $\eta p^2$ = .02 | 1.42, p = .24, $\eta p^2$<.01 |
| Output Demands | 6.53 (10.27) | 14.62 (21.17) | 9.36 (13.62) | 10.45 (16.86) | 0.07, p = .879, $\eta p^2$<.01 | **3.95, p<.05, $\eta p^2$ = .02**\* | 2.31, p = .13, $\eta p^2$ = .01 |

[a]n = 52,

[b]n = 40,

[c]n = 45,

[d]n = 38 as only participants in employment completed this measure.

[e]higher score indicates better functioning.

\* p<.05.

\*\* p<.01.

$\eta p^2$ = partial eta squared.

met the DSM-IV symptom criteria than in the comparison group ($X^2$ (1) = 10.63, $p$ < .01, V = .27). In the mTBI group, a significantly greater proportion of women (small to medium effect size) met the cut-off scores compared to the men ($X^2$ (1) = 5.22, $p$ = .02, V = .19), but there was no significant association, with a small effect size, in the comparison group ($X^2$ (1) = 0.75, $p$ = .387, V = .07). Over a third (36%, n = 54, 27 men and 27 women) of the participants with mTBI reported that they thought they were still affected by the brain injury they had 8-years ago.

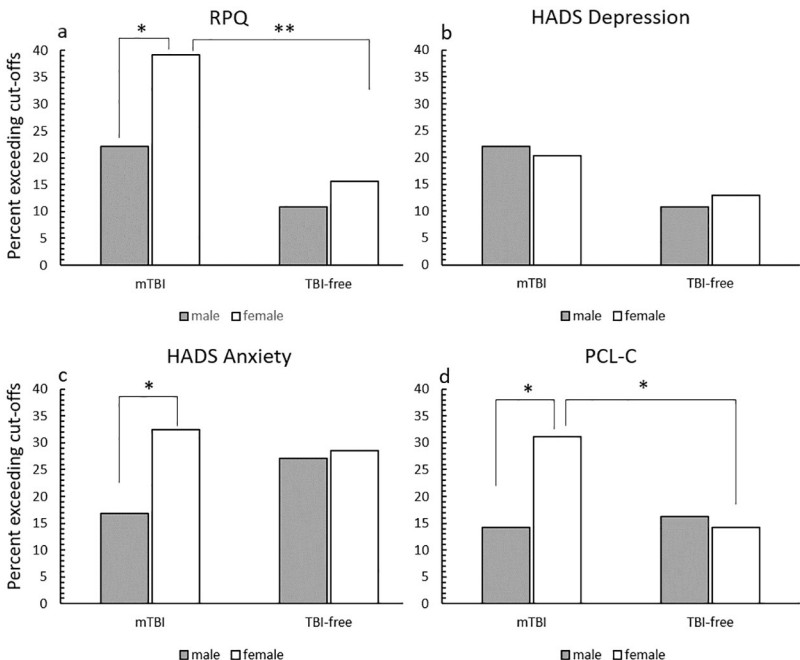

**Fig 2. The percentage of participants in the mTBI and comparison group meeting the cut-off scores for the RPQ (a), HADS depression (b), HADS anxiety (c) and PCL-C (d) Note: Data are presented as the percentage of the group meeting the cut-off score.** N for mTBI group = 151 (males = 77, females = 74); N for TBI free group = 151 (males = 74, females = 77). The cut-off for the RPQ was 4 or more items with a rating >2; the cut-off score for the HADS was ≥8 and the cut-off score for the PCL-C was ≥14. * p<.05 **p<.01 from chi-square tests.

The ANOVAs for the item level analyses are presented in S1 Table. One item on the RPQ, blurred vision, showed a significant Group x Sex interaction ($p$ = .003, with a small to medium effect size); women with mTBI reported significantly higher symptom severity than the other three groups. The mTBI group reported significantly increased symptoms on 13 of the RPQ items (headaches, dizziness, nausea, noise sensitivity, irritable, frustrated, forgetfulness, concentration, longer to think, blurred vision, light sensitivity, double vision, restlessness, $ps \leq$ .003) compared to the TBI-free group. Dizziness, forgetfulness and difficulty thinking showed the largest effects ($\eta p^2 \geq$ .09), concentration, blurred vision and restlessness had medium effect sizes ($\eta p^2 \geq$ .05). Women gave significantly higher ratings, with small to medium effect sizes, for five RPQ items (headaches, dizziness, noise sensitivity, fatigue, and blurred vision, $ps \leq$ .002) compared to men, regardless of group.

The mean scores for the HADS and the PCL-C were below clinical cut-offs (8 for the HADS, 14 for the PCL-C; Table 3). The mTBI group obtained significantly higher scores on the depression scale than the TBI-free group, but the effect was small. A significantly greater proportion of the mTBI group (n = 32, 21.2%) obtained scores over the clinical cut-off for the depression scale compared to the TBI-free group, with a small to medium effect (n = 18 (11.9%); $X^2(1)$ = 4.70, $p$ = .03, V = .13). The proportions of men and women exceeding the cut-offs in the mTBI ($X^2(1)$ = 0.07, $p$ = .79, V = .02) and TBI free groups ($X^2(1)$ = 0.17, $p$ = .68, V = .03) were not significantly different (Fig 2b). The proportion of men with scores of 8 and over in the mTBI and comparison group were not significantly different ($X^2(1)$ = 3.47, $p$ = .06, V = .15), although the effect size was in the small to medium range; the proportion of women did not differ significantly between the groups either ($X^2(1)$ = 1.45 $p$ = .23, V = .10).

Women reported significantly greater anxiety than men regardless of mTBI status, but the effect size was small. There was no significant difference in the overall proportion of participants obtaining scores above the clinical cut-off for anxiety between the mTBI (n = 37, 24.5%) and comparison group (n = 42, 27.8%; $X^2(1) = 0.43$, $p = .51$, V = .04), see Fig 2c. The proportion of males and females obtaining scores above the clinical cut-off did not differ significantly between the groups (males $X^2(1) = 2.27$, $p = .13$, V = .12; females $X^2(1) = 0.27$, $p = .61$, V = .04). There was no significant difference in the proportion of males and females exceeding the cut-off score in the TBI-free group ($X^2(1) = 0.45$, $p = .83$, V = .02). In the mTBI group, a significantly greater proportion of females obtained scores above the cut-off than males with a small to medium effect ($X^2(1) = 4.93$, $p = .03$, V = .18). The item level analysis for the anxiety scale revealed that women reported significantly more severe symptoms than men for one item, feelings of panic ($p < .001$), with a small to medium effect size.

Scores on the PCL-C showed a significant Group x Sex interaction, but the effect was only small; women with mTBI obtained significantly higher scores than men with mTBI and women in the comparison group ($ps < .05$). In addition, there were significant main effects of group (the mTBI group had higher scores than the TBI-free group) and sex (females had higher scores than males), both with small effect sizes. There was no significant difference in the proportion of participants in the mTBI (n = 34, 22.5%) and comparison group (n = 23, 15.2%) scoring over the PCL-C cut-off ($X^2(1) = 2.62$, $p = .11$, V = .09), Fig 2d. In the mTBI group a significantly greater proportion of women compared to men scored above the cut-off, a small to medium effect ($X^2(1) = 6.10$, $p = .014$, V = .20), Fig 2d. There was no sex difference in the comparison group ($X^2(1) = 0.11$, $p = .74$, V = .03). A significantly greater proportion (small to medium effect) of women in the mTBI group scored over the cut-off compared to the TBI-free group ($X^2(1) = 6.10$, $p = .014$, V = .20); there was no significant association for males ($X^2(1) = .11$, $p = .74$, V = .03). The item level analysis revealed a main effect of group for one item, difficulty concentrating ($p = .001$). The mTBI group obtained higher scores than the TBI-free group, but the effect size was small.

## Functional status

The scores on the community participation measure were similar for all groups except for the social relations subscale, where the mTBI group obtained a significantly higher score than the comparison group, indicating greater engagement in social activities. The effect sizes were small for all analyses (Table 3).

On average, in terms of work outcomes (WLQ), participants' health impacted their ability to work less than 20% of the time (Table 3). The Time Demands subscale showed a significant Group x Sex interaction. Women with mTBI reported that their health had a significantly greater impact on this aspect of their work than the men with mTBI ($p < .05$) with a small to medium effect size. The mTBI group reported that their health had a significantly greater impact on the physical demands of their job compared to the TBI-free group. Overall, women reported that their health impacted the output demands of their job significantly more than men, regardless of their TBI status. However, the effect was small in both cases. There were no statistically significant effects from the item level analyses, and the effect sizes were small (S1 Table).

Interestingly, 12.7% of men and 13.6% of women in the mTBI group who had changed their job reported that they did so because of their injury. In addition, 17.6% of men and 15.7% of women reported having to make changes to their job to enable them to continue in their roles since their injury.

**Table 4. Descriptive and inferential statistics for males and female with single or repetitive TBI for each outcome measures.** Data are presented as mean (standard deviation) for males and females with a single or repetitive TBI eight years post-injury for measures of symptom and symptom burden and functional outcomes. Two-way ANOVAs were conducted to explore differences by group and sex.

| | Single TBI M (SD) n = 61 | | Repetitive TBI M (SD) n = 90 | | Two way ANOVA | | |
|---|---|---|---|---|---|---|---|
| | Male (n = 24) | Female (n = 37) | Male (n = 53) | Female (n = 37) | Group (F, sig, partial eta square) | Sex (F, sig, partial eta square) | Group x Sex (F, sig, partial eta square) |
| **Post-Concussion Symptoms (RPQ)** (df = 1, 147) | | | | | | | |
| Total | 11.17 (12.12) | 15.03 (14.47) | 15.28 (13.18) | 26.05 (16.72) | **9.80, p<.01, $\eta p^2$ = .06**[**] | **9.15, p<.01, $\eta p^2$ = .06**[**] | 2.04, p = .16, $\eta p^2$ = .01 |
| Cognitive | 2.75 (3.19) | 3.30 (3.72) | 3.89 (3.63) | 5.76 (3.72) | **8.66, p<.01, $\eta p^2$ = .06**[**] | 3.91, p = .05, $\eta p^2$ = .03 | 1.17, p = .28, $\eta p^2$<.01 |
| Somatic | 3.17 (4.30) | 5.42 (5.49) | 5.15 (4.92) | 10.11 (7.29) | **12.26, p<.01, $\eta p^2$ < .08**[**] | **14.19, p<.01, $\eta p^2$ = .09**[**] | 2.03, p = .16, $\eta p^2$ = .01 |
| Emotional | 3.71 (4.47) | 3.78 (4.06) | 3.89 (4.13) | 6.08 (4.49) | 3.01, p = .09, $\eta p^2$ = .02 | 2.53, p = .11, $\eta p^2$ = .02 | 2.20, p = .14, $\eta p^2$ = .01 |
| **Mood (HADS and PCL)** (df = 1, 147) | | | | | | | |
| Depression | 4.29 (4.56) | 3.68 (3.60) | 3.79 (3.59) | 5.65 (4.43) | 1.20, p = .28, $\eta p^2$ = .01 | 0.85, p = .36, $\eta p^2$<.01 | 3.38, p = .07, $\eta p^2$ = .02 |
| Anxiety | 3.52 (2.75) | 5.11 (4.01) | 5.04 (3.94) | 7.00 (4.64) | **6.24, p = .01, $\eta p^2$ = .04**[*] | **6.77, p = .01, $\eta p^2$ = .05**[*] | 0.08, p = .78 $\eta p^2$<.01 |
| PTSD | 8.67 (2.88) | 10.05 (4.65) | 10.11 (4.38) | 13.51 (6.26) | **9.15, p<.01, $\eta p^2$ = .06**[**] | **8.71, p<.01, $\eta p^2$ = .06**[**] | 1.54, p = .22, $\eta p^2$ = .01 |
| **Community Participation (Part-O)** (df = 1, 146)[e] | | | | | | | |
| Productivity | 2.20 (0.86) | 2.21 (1.02) | 2.09 (0.98) | 2.10 (1.29) | 0.36, p = .55, $\eta p^2$<.01 | 0.01, p = .95, $\eta p^2$<.01 | 0.01, p = .98 $\eta p^2$<.01 |
| Social Relations | 2.37 (1.05) | 2.58 (0.90) | 2.62 (1.00) | 2.60 (0.90) | 0.68, p = .41, $\eta p^2$<.01 | 0.35, p = .55, $\eta p^2$<.01 | 0.53, p = .47, $\eta p^2$<.01 |
| Out and About | 1.19 (0.54) | 1.37 (0.59) | 1.38 (0.65) | 1.11 (0.70) | 0.11, p = .74, $\eta p^2$<.01 | 0.18, p = .68, $\eta p^2$<.01 | **4.30, p = .04, $\eta p^2$ = .03**[*] |
| **Work Limitations (WLQ)** (df = 1, 88) | | | | | | | |
| Time Demands | 11.20 (12.98)[a] | 17.28 (17.67)[b] | 7.84 (8.60)[c] | 19.71 (25.29)[d] | 0.18, p = .89, $\eta p^2$<.01 | **6.62, p = .01, $\eta p^2$ = .07**[*] | 0.67, p = .41, $\eta p^2$<.01 |
| Physical | 11.71 (20.22) | 16.67 (23.34) | 11.83 (23.09) | 22.06 (29.01) | 0.28, p = .60, $\eta p^2$<.01 | 2.13, p = .15, $\eta p^2$ = .02 | 0.26, p = .61, $\eta p^2$<.01 |
| Ment/Inter-personal | 10.27 (15.23) | 14.61 (15.26) | 9.21 (8.69) | 17.65 (18.53) | 0.11, p = .74, $\eta p^2$<.01 | **4.51, p = .04, $\eta p^2$ = .05**[*] | 0.46, p = .50, $\eta p^2$<.01 |
| Output | 7.81 (13.90) | 14.24 (22.68) | 5.81 (8.29) | 15.15 (19.61) | 0.02, p = .88, $\eta p^2$<.01 | **5.05, p = .03, $\eta p^2$ = .05**[*] | 0.17, p = .68, $\eta p^2$<.01 |

[a]n = 16,

[b]n = 23,

[c]n = 36,

[d]n = 17 as only participants in employment completed this measure.

[e]higher score indicates better functioning.

[*] p<.05.

[**] p<.01.

$\eta p^2$ = partial eta squared.

### Outcomes from single or repetitive injuries

A series of 2 (Group: single/repetitive) x 2 Sex: male/female) ANOVAs were conducted to examine the effect of multiple TBI on the outcome measures. These data are summarised in Table 4. The group with repetitive TBI reported significantly more symptoms/greater symptom burden with medium effect sizes on the RPQ total, cognitive and somatic subscales than those in the single TBI group. Women had significantly higher RPQ total and somatic subscale scores than men, with medium effect sizes. A significantly greater proportion (small to medium effect) of those with repetitive TBI met the DSM-IV symptom criteria for PCS (n = 33, 36.7%) compared to the single injury group (n = 13; 21.3%; $X^2$(1) = 4.05, p = .04, V = .17). As shown in Fig 3a, the percentage of males above the cut-off did not differ between the groups ($X^2$ (1) = 0.59, p = .44, V = .09), but there was a significantly greater proportion of females with repetitive TBI above the cut-off compared to those with single TBI, a medium effect ($X^2$ (1) = 6.90, p = .01, V = .31). There was no significant association between the

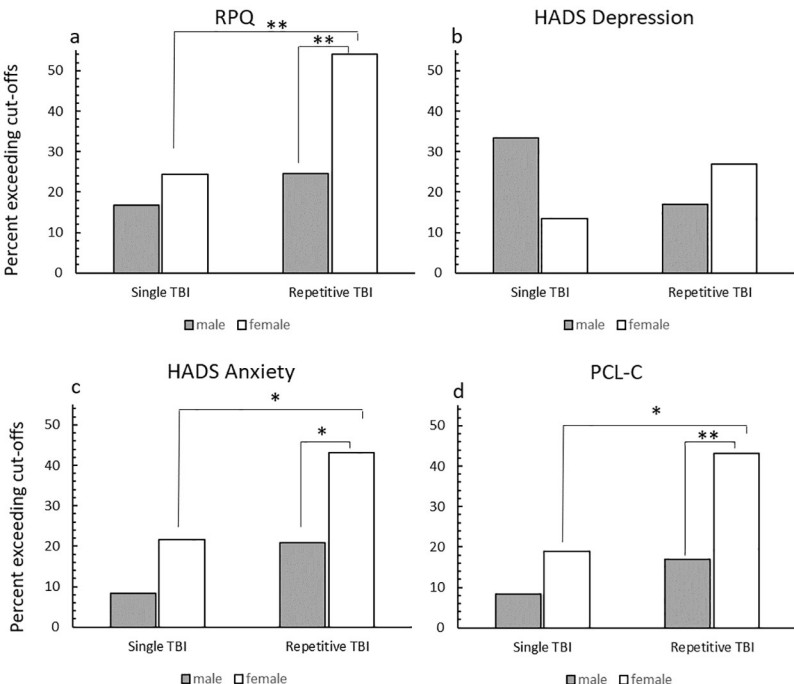

**Fig 3. The percentage of participants in the single and repetitive TBI groups meeting the cut-off scores for the RPQ, HADS depression, HADS anxiety and PCL-C.** Note: Data are presented as the percentage of the group meeting the cut-off score. N for single TBI group = 61 (males = 24, females = 37); N for repetitive TBI group = 90 (males = 53, females = 37) The cut-off for the RPQ was 4 or more items with a rating >2; the cut-off score for the HADS was ≥8 and the cut-off score for the PCL-C was ≥14. * p<.05 **p<.01 from chi-square tests.

proportion of those meeting the RPQ cut-off and sex in the single TBI group ($X^2(1) = .51$, $p = .48$, V = .09), however, in the repetitive TBI group, a significantly greater percentage of females (medium effect size) were above the cut-off compared to males ($X^2(1) = 8.18$, $p < .01$, V = .30).

The repetitive TBI group obtained significantly higher scores with medium effect sizes on the HADS anxiety scale and PCL-C than the single mTBI group. For both measures, women had significantly higher scores than men (medium effect, Table 4).

Fig 3 shows the percentage of participants in each group obtaining scores above the clinical cut-offs for the HADS Depression scale (b), Anxiety scale (c) and the PCL-C (d). There was no significant association between TBI history (single TBI n = 13, 21.3%; repetitive TBI n = 19, 21.1%) and exceeding the depression cut-off score ($X^2(1) < .01$, $p = .98$, V < .01). The comparisons between males and females within each group did not reach statistical significance (single TBI $X^2(1) = 3.41$, $p = .07$, V = .24; repetitive TBI ($X^2(1) = 1.32$, $p = .25$, V = .12), although there was a medium effect size for the single TBI group. There were no significant associations between TBI history and sex (male $X^2(1) = 2.59$, $p = .11$, V = .18; female $X^2(1) = 2.09$, $p = .15$, V = .17) for the depression scale, with effect sizes in the small to medium range.

For the anxiety scale, 16.4% (n = 10) of the single TBI group exceeded the clinical cut-off compared with 30.0% (n = 27) of the repetitive TBI group ($X^2(1) = 3.64$, $p = .06$, V = .15). The proportion of males did not differ significantly (small to medium effect) between the single and repetitive TBI groups ($X^2(1) = 1.82$, $p = .18$, V = .15), but a significantly greater proportion of females exceeded the cut-offs in the group with repetitive TBI compared to the single TBI group ($X^2(1) = 3.95$, $p = .05$, V = .23) (Fig 3c). In the single TBI group the proportion of males and females exceeding the clinical cut-off for anxiety were not significantly different with a small to medium effect size ($X^2(1) = 1.88$, $p = .17$, V = .18). A significantly greater proportion

of females exceeded the clinical cut-off in the repetitive TBI group compared to males, a medium-size effect ($X^2$(1) = 5.25, $p$ = .02, V = .24).

For the PCL-C, there was no significant difference in the proportion of participants exceeding the clinical screening score between the single (8%, n = 9) and repetitive TBI groups (27.8%, n = 25; $X^2$(1) = 3.54, $p$ = .06, V = .15). Similarly, there was no significant difference in the proportion of males meeting the cut-off ($X^2$(1) = 1.01, $p$ = .32, V = .11) between the two groups. A significantly greater proportion of females exceeded the cut-off in the repetitive TBI group ($X^2$(1) = 5.11, $p$ = .02, V = .26) compared to the single TBI group, a medium effect (Fig 3d). Within the groups, there was no statistically significant association between gender and exceeding the PCL-C cut-off in the single TBI group ($X^2$(1) = 1.30, $p$ = .26, V = .15) however, a significantly greater proportion of females (medium effect size) exceeded the cut-off score in the repetitive TBI group compared to males ($X^2$(1) = 7.49, $p$ < .01, V = .29).

There was a significant Group x Sex interaction for the Out and About subscale of the Community Participation scale with a small to medium effect size for the functional outcome measure. Women in the single TBI group and men in the repetitive TBI group reported the highest levels of community participation (Table 4). There was no significant effect of prior TBI on work-related outcomes (WLQ), and the effect sizes were small. Women reported that their health negatively affected their work significantly more than men for the Time Demands, Mental/Interpersonal Demands and Output Demands subscales (medium effect sizes).

## Discussion

The overall aim of the study was to explore sex differences in outcomes from mTBI in men and women at eight years post-injury by comparison with an age and sex-matched TBI-free sample. In addition, we explored sex differences in outcomes from single and repetitive TBI.

The comparisons between the mTBI and TBI-free group revealed that the mTBI group reported greater post-concussion symptoms, depression symptoms, a greater impact of their health on physical work demands and, surprisingly, significantly higher levels of community participation compared to the TBI-free group. Regarding sex differences, women reported more post-concussive symptoms, higher levels of anxiety and a greater impact of their health on work productivity than men. In addition, women with mTBI obtained the highest scores on the PTSD scale and reported that their health had the greatest impact on time-related work demands. Over twice as many females with mTBI met PCS cut-offs compared to any other group (40% compared to <20% for all other groups). In addition, a greater proportion of women with mTBI met the cut-offs for anxiety and exceeded the clinical screening score for PTSD, compared to males with mTBI and females in the comparison group. There was no evidence of interactions between sex and age at injury for the key outcome variables, but this may be due to the limited sample size.

Our sample was identified as part of an epidemiological study to determine TBI incidence, and almost two thirds had experienced a prior TBI. Those with repetitive injuries had poorer outcomes, reporting more post-concussive, anxiety and PTSD symptoms, but the higher symptom burden did not affect work. Within the TBI group, females obtained significantly higher scores (and a greater proportion met the cut-offs) for post-concussive, anxiety and PTSD symptoms compared to males.

Together these findings suggest that those with mTBI show heightened post-concussion, depression and PTSD symptoms up to 8-years post-injury, which may impact work productivity. Females with mTBI, particularly those who have experienced multiple TBI, had the poorest outcomes across a number of domains as shown by the significant proportion meeting the symptom cut-off scores for PCS and PTSD.

This study provides further evidence that the negative impacts of mTBI can persist for a significant period post-injury, and there seems to have been little improvement in PCS symptoms from 4 years post-injury [16]. The proportion meeting the cut-off scores for PCS had decreased from 12 months post-injury (47.9% at 12 months, 30.5% at 8 years), but the proportion meeting the clinical cuts off for the depression scale had increased (12.1% at 12 months, 21.1% at 8 years) and decreased slightly for anxiety (29.3% at 12 months, 24.5% at 8 years) [10]. Female sex was a predictor of poor outcomes for post-concussion symptoms, health-related quality of life- mental component, anxiety and general poor functioning (Glasgow Outcome Scale) at 12 months post-injury, but was no longer a significant predictor 4 years post-injury [10, 16]. Interestingly, only six participants in this cohort had any active TBI-related rehabilitation within 4 years of injury [16]. These findings suggest that symptoms present at 12 months post-injury are unlikely to resolve over a longer period. Some of the sex differences in outcomes may reduce over time, but as a result of attrition, our sample is getting smaller at each follow-up time point, so it is difficult to reach any firm conclusions.

Recovery from mTBI is complex, and there is increasing evidence that psychosocial factors play a key role [33], and careful management of return to work, school and sport can assist with good recovery whilst minimising the risk of subsequent injuries. This cohort was identified in 2010–2011, and at that time, much less was known about the potential adverse consequences of mTBI (or concussion), and as a result, many people did not seek medical attention, and there was limited ongoing monitoring and follow-up. Improvements in the acute management of mTBI and educating people about how best to cope with symptoms and when to seek medical support may help to ameliorate the long-term negative consequences of mTBI.

In general, women reported a higher symptom burden than men regardless of mTBI status, in keeping with previous reports [57, 58]. Our analyses did not provide a clear answer as to whether there were sex differences in long-term outcomes from mTBI. We did not find Group x Sex interactions for PCS, depression or anxiety, but a greater proportion of women with mTBI exceeded the clinical cut-off scores for PCS and PTSD. For PCS, this was a significant proportion of the women with mTBI in the study, 40%, double the proportion of any other group; 30% of the women with mTBI met the screening cut-off for PTSD, twice the proportion of any other group. For women with prior TBI, the proportion exceeding the cut-offs was even higher, over 50% for PCS and over 40% for PTSD. Overall, 24.4% of women with mTBI exceeded the cut-offs on 3 or more measures (PCS, anxiety, depression, PTSD) compared with 7.8% of men with mTBI. This suggests that women may be at greater risk of poor long-term outcomes post mTBI, particularly if they have a history of previous injuries.

Why this group have the poorest outcomes is unclear. The lack of intervention or treatment for prior injuries may have meant that another injury was sustained before full recovery, leading to a greater symptom burden [59]. Alternatively, undiagnosed psychiatric disorders (anxiety, depression, PTSD) may contribute to the higher symptom burden and longer recovery. Despite the high proportion of the sample exceeding the clinical cut-off scores across several measures, only 14.9% of the women with mTBI reported a formal mental health diagnosis, but this was almost double the proportion of men. Whilst these measures are only screening tools, the findings raise the possibility that there are undiagnosed mental health problems within this population. In the context of poor mental health, mTBI could be viewed as another stressor that exacerbates current symptoms. Thus, addressing mental health symptoms may help break the cycle of ongoing symptom burden.

Although recent studies reported interactions between age and sex with symptom burden after mTBI, we did not find any evidence of poorer outcomes for women within a particular age group [31, 32], albeit our sample size meant that we were not able to examine outcomes in smaller age bands. Given the inconsistent findings across studies, further work to explore

the relationship between age at injury and sex differences in adult outcomes from TBI is warranted.

## Strengths and limitations

The sample represents 17.4% of the original incidence sample (n = 870) and 43.6% of those who had agreed to be contacted for follow-up. These response rates are reasonable given that 341 (39.2%) of the original sample completed an assessment in the 12 months post-injury. The sample completing the assessment at 8-year post-injury contained 10% fewer males than the incidence sample. In addition, the proportion of Māori participants was also 10% lower. Māori (indigenous people of New Zealand) face many social and health inequities and are at greater risk of TBI than other ethnicities [2], and potentially poorer outcomes. Thus, we may have underestimated the impact of mTBI eight years post-injury.

A large amount of missing data for education, occupation, and injury severity (GCS) makes it difficult to draw firm conclusions as to the representativeness of the sample for these factors. One of the most challenging issues for longitudinal research is maintaining contact with participants. Contact and communication options have changed significantly since the incidence study in 2010–2011, with a shift towards mobile phones, email and social media rather than home phone and postal contact. We now routinely request multiple different types of contact information to assist with future contact. In addition, we used WHO criteria to define mTBI, not the American Congress of Rehabilitation, which includes the presence of observable signs and subjective symptoms (dizziness, confusion) [60, 61].

While we had a reasonably good follow-up rate, our sample was relatively small limiting our age analyses to two groups. Previous studies of mTBI have found that middle-aged women (30–49 years) have poorer outcomes as compared to other age groups [32, 33], and therefore, we may have missed any age-related differences in outcomes due to our limited sample size.

There were no significant differences between the mTBI and comparison groups on key demographic and occupational variables. Of note, there were no significant associations between group or sex in occupation level or the proportion who were the primary income earner. However, it would be appropriate to collect general medical information from the comparison sample in future studies to ensure that group differences were not a result of other co-morbid diagnoses. Ideally, we would include an 'other' injury comparison group [33], but this raises questions about what 'other' injury would be suitable. Our sample had mTBI, and the majority received no formal diagnostic testing or active treatment, and some did not seek medical treatment at the time of injury; thus a community comparison group was deemed most appropriate.

We used the RPQ differently from other studies as we asked participants to rate their symptoms over the 24 hours [41] rather than comparing how they feel now to before their injury to improve accuracy. As a result, we removed the response option 1, no more of a problem, as the response did not make sense without a comparison time-point. It should be noted that this did not affect the scores, as the rating of 1 is not included in calculating the total or subscale scores.

Future studies could consider including assessments of malingering, and social desirability or using measures that incorporate measures of response consistency. Finally, the assessment time point (8-years post-injury) was opportunistic (we applied for and received funding), and longer-term follow-up studies are warranted. Despite these limitations, the current study addressed key gaps in the literature by examining the long-term outcomes of mTBI in men and women.

In conclusion, at 8-years post-mTBI, people may experience a higher symptom burden, and women with repeat injuries appear to be at the highest risk of poor outcomes. Providing

education about injury management, return to work, and screening for, and treating, mental health issues in the acute post-injury phase may help to decrease the long-term symptom burden.

## Supporting information

**S1 Table. Descriptive and inferential statistics for the males and females in the mTBI (8-years post-injury) and comparison groups for items on the outcome measures with a significant main effect or interaction of sex RPQ, HADS anxiety, PCL-C, WLQ Time demands, WLQ Output demands).**
(DOCX)

## Acknowledgments

Thank you to the participants for their ongoing commitment to the study and the research assistants involved in the data collection. We would also like to thank Waikato District Health Board, general practitioners and health care providers for their ongoing support.

BIONIC8 Study Group members include Nicola Starkey (lead author, nstarkey@waikato. ac.nz), Brittany Duffy (University of Waikato, Hamilton, New Zealand); Valery Feigin, Alice Theadom, Kelly Jones, Priya Parmar, Amy Jones (AUT University, Auckland, New Zealand); Suzanne Barker-Collo, Shanthi Ameratunga (University of Auckland, Auckland, New Zealand); Anthony Dowell (University of Otago, Dunedin, New Zealand), Michael Kahan (Waikato Occupational Services, Hamilton, New Zealand), Grant Christey (Waikato District Health Board, Hamilton, New Zealand), Natalie Hardaker (Accident Compensation Corporation, Wellington, New Zealand).

## Author Contributions

**Conceptualization:** Nicola Jayne Starkey, Brittney Duffy, Kelly Jones, Alice Theadom, Suzanne Barker-Collo, Valery Feigin.

**Data curation:** Nicola Jayne Starkey, Kelly Jones, Alice Theadom, Suzanne Barker-Collo.

**Formal analysis:** Nicola Jayne Starkey, Brittney Duffy.

**Funding acquisition:** Nicola Jayne Starkey, Brittney Duffy, Kelly Jones, Alice Theadom, Suzanne Barker-Collo, Valery Feigin.

**Methodology:** Nicola Jayne Starkey, Kelly Jones.

**Project administration:** Nicola Jayne Starkey, Brittney Duffy.

**Resources:** Nicola Jayne Starkey.

**Supervision:** Nicola Jayne Starkey, Valery Feigin.

**Writing – original draft:** Nicola Jayne Starkey, Brittney Duffy.

**Writing – review & editing:** Nicola Jayne Starkey, Brittney Duffy, Kelly Jones, Alice Theadom, Suzanne Barker-Collo, Valery Feigin.

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
