## [Decision Letter · Decision Letter 0]

4 Oct 2021

PONE-D-21-27103Eight-year health, community participation and work outcomes following mild traumatic brain injury in men and women.PLOS ONE

Dear Dr. Starkey,

Thank you for submitting your manuscript to PLOS ONE. After careful consideration, we feel that it has merit but does not meet PLOS ONE’s publication criteria as it currently stands. Therefore, we invite you to submit a revised version of the manuscript that addresses the points raised during the review process.

As you can appreciate from the attached evaluation, the reviewers raised serious concerns about several methodological and analytical weaknesses in your manuscript.  Specific points that need to be addressed before reconsideration for publication are listed below:

Abstract: 

Please include a more concise description of the study design, measurement time points, patient- or external-rated assessments, and clear statements for the three outcome measures. Please also provide a conclusion and a clear statement of the implications of your results.

Introduction:

Please rewrite with focus on mTBI and outcome with respect to neuropsychiatric burdens regarding RPQ, HADS, PTSD as well as community participation and work as well as sex differences. Additional references to studies in general population should be included.

Methods:

The mixed study design: please clarify the data collection and clearly indicate which patient data are retrospective data from a previous study and which part represent the cross-sectional analysis at 8-years after mTBI.

Different sample size: please clearly explain and justify the different sample size for the mTBI and control groups. 

Control group: Please indicate whether the control group matched the mTBI group in terms of educational level and ethnicity. Please provide a more precise definition of mTBI

Inclusion criteria should be specified, as it seems that also repetitive mTBI patients were included, these patients might hamper results and should be excluded, particularly as the pathophysiology of repetitive TBI and single TBI seems to differ according to the literature. Please also justify the inclusion of adolescents aged > 16 years instead of merely adults.

The paragraphs on assessing community participation and work would benefit from further clarity. Maybe I missed something, but it remained unclear whether those parameters were assessed by self-rating or external ratings? Were the applied instruments validated for the assessed age groups? Further information would be helpful.

Explain how the parameter sex/gender was evaluated. Which confounders were assessed to minimize gender factors that might influence the detrimental females’ outcome such as financial inequity, social position, marital status, etc. between sexes?

The description of the statistical analyses needs to include primary and secondary outcome measures. 

Please provide sufficient detail about the participant characteristics including parameters such as GCS, mechanism of injury, prior TBI etc.) in Table 1. In Table 2 please report the characteristics of participants for all subgroups. Statistically insignificant results and associated average results should be reported.

Results and Discussion

In the Results absolute numbers e.g. for the Rivermead Post-Concussion Symptom Questionnaire assessments should be interpreted in terms of the given cut-offs in the literature.

Interpretation of the data should include the possible influence of study population/ characteristics of the included participants, methodological limitations and the literature in the field.  The study limitations should be discussed thoroughly.  Plausible explanations and clinical implications of the results should be also discussed.

Descriptive characteristics e.g. sex and ethnicity should be presented for all subgroups.  

Information on the characteristics of participants who did and did not complete the 8-year measurement should be provided in terms of GCS, mechanism of injury, CT abnormalities.

Report and include in the analysis psychiatric and general medical history and health status of the participants. 

The Rivermead Post-Concussion Symptom Questionnaire was used without the comparison before the injury.  This  unusual reporting should be noted in the Discussion and related to the observed average scores. 

The literature should be updated.

We look forward to receiving your revised manuscript.

Kind regards,

Anna-Leena Sirén

Academic Editor

PLOS ONE

Journal Requirements:

2. Please describe in your methods section how capacity to provide consent was determined for the participants in this study. Please also state whether your ethics committee or IRB approved this consent procedure. If you did not assess capacity to consent please briefly outline why this was not necessary in this case.

4. One of the noted authors is a group or consortium "BIONIC8 Research Group". In addition to naming the author group, please list the individual authors and affiliations within this group in the acknowledgments section of your manuscript. Please also indicate clearly a lead author for this group along with a contact email address.

Reviewers' comments:

Reviewer's Responses to Questions

**Comments to the Author**

1. Is the manuscript technically sound, and do the data support the conclusions?

Reviewer #1: No

Reviewer #2: Partly

2. Has the statistical analysis been performed appropriately and rigorously? 

Reviewer #1: No

Reviewer #2: No

3. Have the authors made all data underlying the findings in their manuscript fully available?

Reviewer #1: No

Reviewer #2: No

4. Is the manuscript presented in an intelligible fashion and written in standard English?

Reviewer #1: No

Reviewer #2: Yes

5. Review Comments to the Author

Reviewer #1: Due to major issues regarding the study design, the methods, results, and the statistical analysis, I made suggestions how the presentation of this important work could be improved for resubmission. Please see the attached file for all my comments. Thanks for your valuable work in the fild of neuropschiatric burdens after mild TBI.

Dear authors,

thank you very much for your scientific effort to elucidate long-term outcome after mild TBI (mTBI), which is an underestimated neuropsychiatric burden, and thus this work will help closing the gap of evidence in the field.

Please see my comments regarding weaknesses and my suggestions that might help to improve clarity of your manuscript.

• The study design needs to be clarified, as it seems to be a mixed design including retrospective data from a previous study (RPQ and HADS from 1- and 12-months after mTBI) and a cross-sectional analysis at 8-years after mTBI. Maybe I missed something, but in the current form it remains somehow unclear or even misleading. It is well known that long-term follow up after TBI is challenging due to immense loss of data/follow up, thus this idea of combining data in principle is good enough but should be clearly stated. As there were only early results at 1- and 12-months post-mTBI available for the RPQ and HADS this should be the primary outcome over time, and thus the study design should be clarified throughout the title, abstract and whole MS as mentioned above.

• It remains unclear why sample sizes are different (mTBI: N=151, controls: N=211), and it is important to have a control group which does not differ in terms of educational level and ethnicity, both relevant factors in terms of neuropsychiatric burdens after TBI.

• The definition of mTBI should be more precise according to the ACMR criteria with negative CCT scans. If these ACMR criteria were not used, it should be better explained and clarified. Alternatively, definitions which were used in the CENTER-TBI (uncomplicated, complicated mTBI) or TRACK-TBI study could be used and information on CCT/cMRI would be desirable.

• Mild TBI should be used as mTBI instead MTBI

• There is a need of explanation on the rationale for including adolescents aged > 16 years instead of merely adults. Were the 16-year-olds from the school or working population at the time of their TBI? Sometimes inclusion was > 16, sometimes ≥ 16 years; this should be adopted. However, I recommend including adults ≥ 18 years of age at TBI.

• Inclusion criteria should be specified, as it seems that also repetitive mTBI patients were included, these patients might hamper results and should be excluded, particularly as the pathophysiology of repetitive TBI and single TBI seems to differ according to the literature.

• The Methods part in terms of outcome measures should be more precise and I suggest the following organization:

1. Neuropsychiatric burden measured by patient-reported outcome measures (PROMs: RPQ, HADS, PCL-C) instead of using “health”

2. community participation (Participation Assessments with Recombined Tools)

3. work (Work Limitations Questionnaire)

• I suggest using the common wording of PROMs

• All outcome measures should be revised and particularly cut-off scores should be given.

• The paragraphs on assessing community participation and work would benefit from further clarity. Maybe I missed something, but it remained unclear whether those parameters were assessed by self-rating or external ratings? Were the applied instruments validated for the assessed age groups? Further information would be helpful.

• I suggest to be careful with the diagnosis of PTSD as you only assessed PTSD symptoms with the PCL-C checklist.

• The abstract would benefit from more concise description of the study design, measurement time points, patient- or external-rated assessments, and clear statements for the three outcome measures

1. Neuropsychiatric sequels measured by PROMs (RPQ, HADS, PCL-C)

2. community participation (Participation Assessments with Recombined Tools)

3. work (Work Limitations Questionnaire)

• The introduction would benefit from a better focus. Overall, the intro should be shorter with focus on mTBI and outcome with respect to neuropsychiatric burdens regarding RPQ, HADS, PTSD as well as community participation and work as well as sex differences; in the current stage the intro is broad and the reader gets somehow lost within lots of information which is not clearly focus on the presented study.

• The methods part is mixed with results (referral to Table 1). The same is relevant for Table 2, which is a result as well. I suggest to merely describe the methods which you used to get the controls. All details about the controls belong to the results part. I think it is a problem that controls and mTBI subjects differed in terms of education and ethnicity as stated above and this should be adopted when possible.

• Table 1 is a result and should be changed in terms of the column order, thereby it would be easier to anticipate which sample was included and which differed significantly (p-values should directly be incorporated). As the analyzed sample is not representative for the initial sample in terms of age, sex distribution and ethics, this is not ideal for data interpretation, and thus results in the current format need to be handled with care and should merely be descriptive. Due to this major limitation, I suggest to get another statical support and recalculate the data by using imputation analysis, or ITT or completer analysis, which might strengthen the conclusions.

• Could you please improve resolution of images?

• Could you please provide an abbreviation list, e.g. for SES p. 5, line 87

• What is the rationale for defined age groups 16-44 and 45 +? This should be clarified.

• Which confounders were assessed to minimize gender factors that might influence the detrimental females’ outcome such as financial inequity, social position, marital status, etc. between sexes?

• Some wording should be double checked, e.g. the “8-year anniversary”, p. 13, line 239

• The stats part needs revision in terms of primary and secondary outcome measures as indicated above. It seems that there were also patients with more than a single mTBI, these should be analyzed separately and not included in this study. Was data normal distributed, descriptive analysis should be given.

• In the result part absolute numbers e.g. for the RPQ should be interpreted in terms of the given cut-offs in the literature. For example, it should be stated whether patients had a manifest post-concussion syndrome. For definitions see e.g. Riemann et al. 2021 (DOI: 10.3171/2020.9.PEDS20421).

• The current literature should be updated, e.g. include references from doi.org/10.1016/S1474-4422(17)30371-X; doi.org/10.1186/s12955-020-01391-3;
doi.org/10.1007/s11357-020-00273-2

• I assume that the authors are English native speakers. However, some parts of the manuscript would benefit from scientific/ language editing aiming a concise scientific English language.

Thank you very much for your immense research effort.

Reviewer #2: This is an interesting article focusing on the long-term consequences of mild TBI in context of sex or gender. Because of the inclusion of a comparison group, long follow-up, inclusion of mild TBI patients from a broad range of settings, and the analysis of the longitudinal trajectory, this study could contribute to the literature in the field.

However, there are some important limitations and concerns related to the analyses, reporting of the results and interpretation/ implications of the results.

Analysis and reporting of the results:

- The analyses do not adequately adjust for differences in ethnicity and education, although the authors state themselves in the Introduction that these differences can be related to outcomes.

- The characteristics of participants are not reported in sufficient detail: characteristics of mTBI patients (GCS, mechanism of injury, prior TBI etc.) should be added to Table 1., so that it is visible how they relate to attrition.

- The characteristics of participants are not reported for all subgroups (e.g. group by sex) being compared (Table 2).

- Statistically insignificant results and associated average results are not reported, as they should be, at least in Supplement.

- It is not clear if other relevant medical history and sociodemographic characteristics were measured in this study: if yes, they should be reported and included in the analyses. If not, that should be discussed in detail as limitation.

- Sex or gender is very important in this study but it is not explained how exactly was measured.

Discussion and conclusion

- The results are not sufficiently interpreted in regard to study population/ characteristics of the included participants, methodological limitations and the literature in the field. The study limitations should be discussed thoroughly. Plausible explanations and clinical implications of the results should be also discussed.

In more detail:

- Abstract: the conclusion and the implications of your results are not clearly stated.

- Introduction: there are couple of other studies in general population, and a recent study in TRACK-TBI data that does include a comparison group and could be mentioned.

- I suggest finding a different wording instead of “mood outcomes” .

- The authors talk about men and women, but about “sex” and not “gender”. How was this variable measured and how is the term understood by the authors?

- There are differences in education and ethnicity. “The MTBI and Comparison group differed in terms of ethnicity and education, but as these were true group differences it was deemed inappropriate to include these variables as covariates in the analyses.(48, 49)” – But it is stated (correctly) in the Introduction that “a range of factors including age, minority ethnicity, lower levels of education, and SESare linked to poorer outcomes” . Therefore, if you would like to compare outcomes in groups with history/no history of TBI, they should be adjusted based on these factors (ethnicity, education), in the way you are incorporating age. Otherwise, you cannot know if the (lack of) difference is due to TBI status or differences in education and ethnicity. In addition, considering the recruitment and data collection process, how can you know that “these were the true group differences”?

- Descriptive characteristics e.g. sex and ethnicity are not presented for subgroups you analyze ( separately for women with/without TBI, for men with/without TBI ). In addition, are there other available information for the groups?

- What are the characteristics of participants who did and did not complete the 8-year measurement in terms of GCS, mechanism of injury, CT abnormalities?

- Reporting results based on p-values is inappropriate. I would like to see average values for all compared subgroups at least in Supplement.

- Why the comparison “TBI-free group” was not compared with the 1-month and 1-year results of the mTBI group?

- The RPQ was used differently (without the comparison before the injury) than in other TBI studies- that should be noted in the Discussion and related to the observed average scores.

- Is it possible to report and include in the analysis other very important factors: psychiatric and general medical history/ health status? If not, this should be discussed in detail as a limitation. How could that influence the results and comparison between groups?

- The average age is relatively low, and the percentage of prior mTBI and the percentage of mechanical force-induced injuries is high. Does it mean that the study included substantial percentage of sport or work- related injuries in young adults/adolescents? How could this influence the results in the context of age?

- It is not discussed how the time could influence the results- e.g. maybe sex x group differences exist 3-6 months after injury but disappear with time (maybe not)? Moreover, discuss why you would expect symptoms after mild TBI so long as 8 years after injury.

- How the patterns can be explained? Why women with mild TBI showed more PTSD symptoms and time-related work demands? How come that symptoms do not change from 1 year to 8 year? To what degree could that be related to the characteristics of responders?

- Age comparisons (age subgroups)- 8 years is a long time. What is more important, the age of injury or 8 years after, considering relevant biological and social factors?

- What are the implications of the results for clinical practice and patients with mild TBI?

6. PLOS authors have the option to publish the peer review history of their article (what does this mean?). If published, this will include your full peer review and any attached files.

Reviewer #1: **Yes: **Dr. Katrin Rauen, FEBN

Reviewer #2: No

---

## [Author Response · Author response to Decision Letter 0]

14 Jan 2022

15th December 2021

Dear Editor and Reviewers,

Thank you for your comprehensive and thoughtful feedback on our manuscript. We have endeavoured to address all of your comments. We have provided line and page numbers to point the reviewers to the revised sections of the manuscript where appropriate. It should be noted that we have made significant changes to the manuscript so the tracked changes version may be rather difficult to follow.

At the outset you might find it helpful to know that we have removed the longitudinal analyses and the paper now focuses on outcomes at eight years post-injury. We have also added some additional analyses comparing participants with their first versus repeat TBI (in response to one of the reviewers’ comments).

The data is available on request from the University of Waikato Ethics committee (email: humanethics@waikato.ac.nz). (Please note that because the research required review by the National Health and Disability Ethics Committee (HDEC), the University Committee did not review the application as the national committee supersedes them; they will however be the contact point for data access as this service is not provided by the national committee). The data cannot be shared because the participants did not provide consent for their data to be shared (even in a de-identified form). The ethics approval for the study and consent process stated that data would not be shared beyond the study team. In addition, New Zealand is a small country and participants are drawn from a region within NZ. Our dataset contains a lot of personal details and medical information, including information about date, time and place of injury, medical provider and subsequent illnesses and injuries. The level of detail in the dataset means that the participants are potentially identifiable by the amount and nature of the data we hold. 

We have provided affiliations for every member of the BIONIC8 research group and contact details of the lead author in the acknowledgements section. Finally, we have uploaded the supplementary table as supporting information.

Editors comments.

Abstract: 

1. Please include a more concise description of the study design, measurement time points, patient- or external-rated assessments, and clear statements for the three outcome measures. Please also provide a conclusion and a clear statement of the implications of your results.

We have provided a clear statement of the research questions, the assessment time point, and that all measures were self-report. We have also provided a clear statement of the implications of the findings

Introduction:

2. Please rewrite with focus on mTBI and outcome with respect to neuropsychiatric burdens regarding RPQ, HADS, PTSD as well as community participation and work as well as sex differences. Additional references to studies in general population should be included.

We have re-organised the introduction to describe previous research describing outcomes from mTBI for the measures we used (ie, post-concussive symptoms, anxiety, depression, PTSD, participation and work outcomes). We have added more recent references that incorporate general population samples and several more recent studies examining the interaction between sex and age on outcomes from mTBI. We have removed the more general information that was not of relevance to the study.

Methods:

3. The mixed study design: please clarify the data collection and clearly indicate which patient data are retrospective data from a previous study and which part represent the cross-sectional analysis at 8-years after mTBI.

We have removed the longitudinal analysis and the paper focuses on the data at eight years post-injury. The injury related information and demographic information presented in Table 1 were collected as part of the original incidence study. The remaining data (presented in Table 2 onwards) was collected as part of the 8-year outcome study.

4. Different sample size: please clearly explain and justify the different sample size for the mTBI and control groups. 

We initially recruited a comparison group matched on age and sex (on average rather than one to one matching) which was larger than the mTBI cohort. In light of the reviewers comments we conducted additional matching of the groups on occupation and reduced the number of participants in the comparison group to 151 (the same as the mTBI group) 

5. Control group: Please indicate whether the control group matched the mTBI group in terms of educational level and ethnicity. 

The smaller comparison group now matches the mTBI group in terms of educational levels and ethnicity as shown in Table 2. A significantly smaller proportion of women in the mTBI and comparison group had completed Trade or technical certificates compared to the men in both groups

6. Please provide a more precise definition of mTBI

We have provided further explanation of the definition of mTBI cases as shown below. The study took place prior to the CENTER TBI and TRACK TBI studies and used the most suitable criteria available at the time (page 8 line 158 onwards):

“mTBI was defined as Glasgow Coma Scale score of 13-15 and/or post-traumatic amnesia (PTA) of less than 24 hours (39). Each mTBI case was further classified as having low, medium or high risk of developing complications based on the presence or absence of clinical findings (loss of consciousness, amnesia, vomiting or diffuse headache), neurological deficits, (impaired vision, hearing, speech, balance, walking difficulties or weakness), skull fracture and other risk factors (coagulopathy, drug/alcohol consumption, previous neurosurgical procedures, pre-trauma epilepsy, over 60 years of age) (40). mTBI cases with no clinical or neurological deficits, no skull fracture or risk factors were classified as low risk. Those with clinical findings but no neurological deficits, skull fracture or risk factors were classed as medium risk. The high-risk group included those with clinical, neurological, skull fracture and/or other risk factors.”

7. Inclusion criteria should be specified, as it seems that also repetitive mTBI patients were included, these patients might hamper results and should be excluded, particularly as the pathophysiology of repetitive TBI and single TBI seems to differ according to the literature. 

The mTBI cohort was originally identified as part of a population-based epidemiological study of TBI and included those with and without prior TBI. As the overall aim of the study was to explore the symptom burden, health and functional outcomes of this group in comparison to a TBI-free group 8 years post-injury, we have retained the analyses for the whole group, as we think the findings provide useful insights into recovery 8 years after injury, regardless of prior TBI status (which is often not documented or known by clinicians when people seek treatment).

We also acknowledge that those with single or repeat injuries may show different outcomes. To address this, we have conducted additional analyses comparing the 8-year outcomes for men and women who had either single or repeat injuries. These findings are reported in Table 4.

8. Please also justify the inclusion of adolescents aged > 16 years instead of merely adults.

The age of consent in New Zealand is 16 years of age, so we included all of the participants who were able to provide their own consent for the original incidence study. All participants aged 16 years or over completed the same assessments. The current analyse focus on the participants 8 years post-injury so the minimum age of the sample is 24 years.

9. The paragraphs on assessing community participation and work would benefit from further clarity. Maybe I missed something, but it remained unclear whether those parameters were assessed by self-rating or external ratings? Were the applied instruments validated for the assessed age groups? Further information would be helpful.

We have added additional information about the measures in the Measures section. All were self-report measures and validated for this age group. (p12 line 230 onwards) 

10. Explain how the parameter sex/gender was evaluated. Which confounders were assessed to minimize gender factors that might influence the detrimental females’ outcome such as financial inequity, social position, marital status, etc. between sexes?

In the BIONIC incidence and outcome study sex/gender was determined from medical records or self-report by the participant (depending on how the case was located). We have carried out additional analyses comparing males and females in the mTBI and control group for level of education, occupation, and if they are the main income earner (see Table 2). There were no significant associations between gender, group and these variables. In addition, we also explored if the participants reported having enough money to live on (not reported in the manuscript for brevity), and there were no significant associations between group, gender and response to this question. 

11. The description of the statistical analyses needs to include primary and secondary outcome measures. 

The study was observational and aimed to provide a broad overview of long-term outcomes following mTBI. As this wasn’t a clinical trial we did not have primary and secondary outcomes measures. We have reworded the research questions to clarify the primary and secondary aims of the study for clarity (page 8 line 143 onwards): 

“The primary aim was to: 1) Compare symptoms and symptom burden, health-related behaviour, and functional outcomes in males and females with mTBI with an age- and sex-matched TBI-free comparison group. Secondary aims were to: 1) Compare outcomes at 8 years post-injury in males and females aged <45 years and >45 years (at injury) and 2) Compare outcomes in those with first in lifetime and repeat TBI.”

12. Please provide sufficient detail about the participant characteristics including parameters such as GCS, mechanism of injury, prior TBI etc.) in Table 1. 

The revised Table 1 compares the characteristics of the original BIONIC study incidence sample (n=870) to the group completing the 8-year outcome assessment (n=151) to determine the representativeness of the sample. We have included additional demographic (education, occupation) and injury-related information (mechanism of injury, mTBI severity, number of prior TBI, GCS, number with CT scan) in the Table. Please note that there is a missing data for some of the variables for the incidence sample (prior TBI, education, occupation) as this information was obtained via interview, rather than from medical records, and we were not able to interview all of the incidence cohort.

13. In Table 2 please report the characteristics of participants for all subgroups. Statistically insignificant results and associated average results should be reported.

We have provided descriptive statistics for all subgroups (male/female; mTBI/comparison), and included the relevant inferential statistics in Table 2. The descriptive and inferential statistics for the item level analysis have been included in a supplementary table (Table S1)

Results and Discussion

14. In the Results absolute numbers e.g. for the Rivermead Post-Concussion Symptom Questionnaire assessments should be interpreted in terms of the given cut-offs in the literature.

We have added information about cut-off scores that meet the symptom criteria for DSM-IV post-concussion syndrome in the measures section, and provided further explanation of the percentage of participants in each group that meet the cut-offs in the relevant sections of the results and in Figs 2 and 3. 

15. Interpretation of the data should include the possible influence of study population/ characteristics of the included participants, methodological limitations and the literature in the field. The study limitations should be discussed thoroughly. Plausible explanations and clinical implications of the results should be also discussed

We have added a strengths and limitations section in the discussion that includes discussion of the study population and how it compares to those identified in the original incidence study. This also includes discussion of our modifications to the RPQ, the suitability of the TBI free comparison group and the use of self-report assessments (p32 lines 549-578).

We have compared the current findings with outcomes at earlier time points, and it seems that symptoms have not improved much from 12 months post-injury. This suggests that early intervention is the key to preventing long-term symptom burden. Of note only 6 participants in our 4 year follow up reported any type of treatment for their injury (beyond the first medical presentation for the injury). (page 30 line 501 onwards). 

Females with repeat mTBI appear most at risk for ongoing symptoms burden (in terms of the proportion exceeding the clinical cut-offs for the RPQ, HADS and PCL-C). The proportion of participants exceeding the clinical cut-offs for the depression/anxiety/PTSD scales far exceeds the number reporting a diagnosed psychiatric condition, suggesting that there may be a significant number of people with undiagnosed mental health issues. Given the complex interplay of psychosocial factors in post-concussion symptoms as well as depression and anxiety, early intervention after concussion to ensure that symptoms have fully resolved may be key to preventing long-term symptom burden. 

16. Descriptive characteristics e.g. sex and ethnicity should be presented for all subgroups. 

This information has been added to Table 2

17. Information on the characteristics of participants who did and did not complete the 8-year measurement should be provided in terms of GCS, mechanism of injury, CT abnormalities.

This information has been added to Table 1. Please note that there is a significant amount of missing data for the incidence sample for some of the variables (e.g., GCS) as this was often not recorded in the medical records, or participants had not sought medical attention soon after injury. As shown in the Table only a small proportion of the sample had a CT scan, and any CT abnormalities were considered when classifying the mTBI injury severity as low, medium or high risk.

18. Report and include in the analysis psychiatric and general medical history and health status of the participants. 

Information about current and previous health issues (reported at 8 years post-injury) has been added to Table 2 for the participants with mTBI. There were no significant associations between health conditions and sex in the mTBI group. We did not ask participants in the comparison group for information about their current /previous health diagnoses.

19. Rivermead Post-Concussion Symptom Questionnaire was used without the comparison before the injury. This unusual reporting should be noted in the Discussion and related to the observed average scores. 

We have provided further information about scoring and our use of the RPQ in the measures section and raised it as a limitation in the Discussion. We excluded the rating of 1, no more of a problem, because the response option did not make sense when participants were asked to rate their symptoms over the last 24 hrs as they had no previous time point to compare with. Previous studies of general population samples have asked participants to provide ratings over a 24-hour period (e.g., Voormolen et al 2019). We used the 24-hour time period because we thought that participants would be unable to accurately rate their symptoms now compared to 8 years ago, and we could use the same question for the mTBI and comparison groups

20. The literature should be updated.

We have carried out another literature search and added more recent references as appropriate (including those suggested by the reviewers and recent publications on sex x age interactions after mTBI from the TRACK-TBI team).

 Reviewer 1 

Due to major issues regarding the study design, the methods, results, and the statistical analysis, I made suggestions how the presentation of this important work could be improved for resubmission. Please see the attached file for all my comments. Thanks for your valuable work in the field of neuropsychiatric burdens after mild TBI.

Thank you for your helpful and constructive feedback

21. The study design needs to be clarified, as it seems to be a mixed design including retrospective data from a previous study (RPQ and HADS from 1- and 12-months after mTBI) and a cross-sectional analysis at 8-years after mTBI. Maybe I missed something, but in the current form it remains somehow unclear or even misleading. It is well known that long-term follow up after TBI is challenging due to immense loss of data/follow up, thus this idea of combining data in principle is good enough but should be clearly stated. As there were only early results at 1- and 12-months post-mTBI available for the RPQ and HADS this should be the primary outcome over time, and thus the study design should be clarified throughout the title, abstract and whole MS as mentioned above.

Apologies for the confusion. The main purpose of the study was to explore sex differences across a broad range of outcomes from mTBI 8 years post-injury. The longitudinal analysis was supplementary and not the main purpose of the paper (we thought the findings made an interesting addition to the paper, and highlighted the importance of including control group data when examining long-term outcomes). We have removed the longitudinal analysis to ensure the aim of the study is clear, and we have incorporated additional analyses examining outcomes for those with and without prior TBI to address one of your later comments. 

22. It remains unclear why sample sizes are different (mTBI: N=151, controls: N=211), and it is important to have a control group which does not differ in terms of educational level and ethnicity, both relevant factors in terms of neuropsychiatric burdens after TBI.

We initially recruited a comparison group matched on age and sex (on average rather than one to one matching) which was larger than the mTBI cohort. In light of the reviewers comments we conducted additional matching of the groups on occupation and reduced the number of participants in the comparison group to 151 (the same as the mTBI group). The smaller comparison group now matches the mTBI group in terms of educational levels and ethnicity as shown in Table 2. A significantly smaller proportion of women in the mTBI and comparison group had completed Trade or technical certificates compared to the men in both groups

23. The definition of mTBI should be more precise according to the ACMR criteria with negative CCT scans. If these ACMR criteria were not used, it should be better explained and clarified. Alternatively, definitions which were used in the CENTER-TBI (uncomplicated, complicated mTBI) or TRACK-TBI study could be used and information on CCT/cMRI would be desirable.

We have provided further explanation of the definition of mTBI cases as shown below. The study took place prior to the CENTER TBI and TRACK TBI studies and used the most suitable criteria available at the time (page 8 line 158 onwards):

“mTBI was defined as Glasgow Coma Scale score of 13-15 and/or post-traumatic amnesia (PTA) of less than 24 hours (39). Each mTBI case was further classified as having low, medium or high risk of developing complications based on the presence or absence of clinical findings (loss of consciousness, amnesia, vomiting or diffuse headache), neurological deficits, (impaired vision, hearing, speech, balance, walking difficulties or weakness), skull fracture and other risk factors (coagulopathy, drug/alcohol consumption, previous neurosurgical procedures, pre-trauma epilepsy, over 60 years of age) (40). mTBI cases with no clinical or neurological deficits, no skull fracture or risk factors were classified as low risk. Those with clinical findings but no neurological deficits, skull fracture or risk factors were classed as medium risk. The high-risk group included those with clinical, neurological, skull fracture and/or other risk factors.”

24. Mild TBI should be used as mTBI instead MTBI

This has been corrected throughout

25. There is a need of explanation on the rationale for including adolescents aged > 16 years instead of merely adults. Were the 16-year-olds from the school or working population at the time of their TBI? Sometimes inclusion was > 16, sometimes ≥ 16 years; this should be adopted. However, I recommend including adults ≥ 18 years of age at TBI.

Participants aged 16 years and over were classed as adults for all of the BIONIC studies (including the original incidence study), as those aged 16 years or over can legally consent to participate in research without parental permission in New Zealand. Those 16 years and over completed the same adult assessments, all of our previous publications from this cohort have been based on this age range. As this assessment was 8 years post-injury the minimum age of the participants was 24 years (maximum 90 years).

26. Inclusion criteria should be specified, as it seems that also repetitive mTBI patients were included, these patients might hamper results and should be excluded, particularly as the pathophysiology of repetitive TBI and single TBI seems to differ according to the literature.

The mTBI cohort was originally identified as part of a population-based epidemiological study of TBI and included those with and without prior TBI. As the overall aim of the study was to explore the symptom burden, health and functional outcomes of this group in comparison to a TBI-free group 8 years post-injury, we have retained the analyses for the whole group, as we think the findings provide useful insights into recovery 8 years after injury, regardless of prior TBI status (which is often not documented or known by clinicians when people seek treatment).

We also acknowledge that those with single or repeat injuries may show different outcomes. To address this, we have conducted additional analyses comparing the 8-year outcomes for men and women who had either single or repeat injuries. These findings are reported in Table 4.

27. The Methods part in terms of outcome measures should be more precise and I suggest the following organization:

1. Neuropsychiatric burden measured by patient-reported outcome measures (PROMs: RPQ, HADS, PCL-C) instead of using “health”

2. community participation (Participation Assessments with Recombined Tools)

3. work (Work Limitations Questionnaire)

• I suggest using the common wording of PROMs

We have reorganised the measures section (p 19 line 192 onwards). We used a TBI screen and the remaining measures could all be classed as PROMS. We have grouped the PROMS under three commonly used categories (as described by Weldring & Smith 2013): symptoms and symptom burden (Rivermead post-concussive symptoms questionnaire), health-related behaviours (Hospital Anxiety and Depression Scale, Posttraumatic Stress Disorder: Civilian Scale) and functional status (Participation with Recombined Tools – Objective, Work Limitations Questionnaire). The description of the TBI screen has been incorporated into the paragraph describing the study specific questions as it is not a main outcome measure.

28. All outcome measures should be revised and particularly cut-off scores should be given.

We have provided further information on the outcome measures and added cut-off scores for the measures where they are available (page 10 line 199 onwards).

29. The paragraphs on assessing community participation and work would benefit from further clarity. Maybe I missed something, but it remained unclear whether those parameters were assessed by self-rating or external ratings? Were the applied instruments validated for the assessed age groups? Further information would be helpful.

All of the data were based on self-report and information about the suitability of the measures, and their development has been added to the measures section. The measures have been validated for this age range and the PART-O was designed for use with a TBI population.

30. I suggest to be careful with the diagnosis of PTSD as you only assessed PTSD symptoms with the PCL-C checklist.

Thank you for pointing this out. We have modified the wording and clarified that the measure was used to screen for the presence and severity of posttraumatic stress symptoms; we have also modified the wording for the HADS, as similarly we were only assessing self-reported symptoms, and it was not a clinical interview.

31. The abstract would benefit from more concise description of the study design, measurement time points, patient- or external-rated assessments, and clear statements for the three outcome measures

1. Neuropsychiatric sequels measured by PROMs (RPQ, HADS, PCL-C)

2. community participation (Participation Assessments with Recombined Tools)

3. work (Work Limitations Questionnaire)

We have restructured the abstract to provide a clearer description of the study design and outcomes 

32. The introduction would benefit from a better focus. Overall, the intro should be shorter with focus on mTBI and outcome with respect to neuropsychiatric burdens regarding RPQ, HADS, PTSD as well as community participation and work as well as sex differences; in the current stage the intro is broad and the reader gets somehow lost within lots of information which is not clearly focus on the presented study.

We have re-organised the introduction to describe previous research describing outcomes from mTBI for the measures we used (ie, post-concussive symptoms, anxiety, depression, PTSD, participation and work outcomes). We have added more recent references that incorporate general population samples and several more recent studies examining the interaction between sex and age on outcomes from mTBI. We have removed the more general information that was not of relevance to the study.

33. The methods part is mixed with results (referral to Table 1). The same is relevant for Table 2, which is a result as well. I suggest to merely describe the methods which you used to get the controls. All details about the controls belong to the results part. I think it is a problem that controls and mTBI subjects differed in terms of education and ethnicity as stated above and this should be adopted when possible.

We have restructured the method as suggested and all data and tables have been moved to the results. We carried out some additional matching between the mTBI and comparison group so that the n was the same and there are no longer significant differences between the groups. 

34. Table 1 is a result and should be changed in terms of the column order, thereby it would be easier to anticipate which sample was included and which differed significantly (p-values should directly be incorporated). As the analyzed sample is not representative for the initial sample in terms of age, sex distribution and ethics, this is not ideal for data interpretation, and thus results in the current format need to be handled with care and should merely be descriptive. Due to this major limitation, I suggest to get another statical support and recalculate the data by using imputation analysis, or ITT or completer analysis, which might strengthen the conclusions.

We have revised Table 1 to compare the original incidence sample with those completing the 8-year outcome study, and added the inferential statistics. We have included additional variables as requested (injury severity, GCS, education etc). The sample completing the 8-year assessment had a higher proportion of females and a greater proportion of those of NZ European ethnicity, but were similar in age. As can be seen in the table many of the other variables had a large amount of missing data for the incidence sample (as they did not participate in the outcome study and we only had information from medical records). We have acknowledged this as a limitation in the discussion. 

We have removed the longitudinal analysis from the paper, so we have not changed the statistical analyses.

35. Could you please improve resolution of images?

The figures have been changed, and we hope the resolution has improved. They are clearer when the image is downloaded rather than viewing it within the pdf

36. Could you please provide an abbreviation list, e.g. for SES p. 5, line 87

Apologies for not writing the term in full. We have checked the manuscript and written each term in full followed by the abbreviation the first time we use it. 

37. What is the rationale for defined age groups 16-44 and 45 +? This should be clarified.

The two age groups were based on the findings from the TRACK TBI study. Due to the limited sample size we were unable to divide the group into smaller age ranges as the group sizes would be too small for the analyses. This has been clarified in the statistical analysis section as follows (p 15 line 295):

“The age groups were selected because previous studies reported poorer outcomes in those aged <45 years (39) and this resulted in two relatively evenly sized groups for the current analysis.”

38. Which confounders were assessed to minimize gender factors that might influence the detrimental females’ outcome such as financial inequity, social position, marital status, etc. between sexes?

We have carried out additional analyses comparing males and females in the mTBI and control group for level of education, occupation, and if they are the main income earner (see Table 2). There were no significant associations between gender, group and these variables. In addition, we also explored if the participants reported having enough money to live on (not reported in the manuscript for brevity), and there were no significant associations between group, gender and response to this question

39. Some wording should be double checked, e.g. the “8-year anniversary”, p. 13, line 239

We have re-read the manuscript and clarified the wording where required e.g. p13 line 266:

“Participants with mTBI who had agreed to follow-up were contacted by phone or email 8 years (plus or minus two months) after their injury”

40. The stats part needs revision in terms of primary and secondary outcome measures as indicated above.

We have revised the statistical analysis section to fully describe all of the analyses, including checking the distribution of the data. As explained earlier we did not identify primary and secondary outcome variables, and we have removed the longitudinal analyses.

41. It seems that there were also patients with more than a single mTBI, these should be analyzed separately and not included in this study. 

The mTBI cohort was originally identified as part of a population-based epidemiological study of TBI and included those with and without prior TBI. As the overall aim of the study was to explore the symptom burden, health and functional outcomes of this group in comparison to a TBI-free group 8 years post-injury, we have retained the analyses for the whole group, as we think the findings provide useful insights into recovery 8 years after injury, regardless of prior TBI status (which is often not documented or known by clinicians when people seek treatment).

We also acknowledge that those with single or repeat injuries may show different outcomes. To address this, we have conducted additional analyses comparing the 8-year outcomes for men and women who had either single or repeat injuries. These findings are reported in Table 4.

42. Was data normal distributed, descriptive analysis should be given.

We have added an explanation to the statistical analysis section about the distribution of the data (p14 line 283-292). Descriptive data has been included in the tables and the figures (the descriptives for the item-based analyses are in the supplementary table).

“Data were imported from Qualtrics into IBM SPSS version 27 and screened for missing values and outliers. Chi square goodness of fit tests were conducted to check the representativeness of the sample participating in the study at 8 years with the original incidence sample. Descriptive statistics, supplemented by chi square or ANOVA were used to compare the demographic characteristics of the males and females in the mTBI and comparison group. Prior to the main analysis we checked the distribution of the data and found that data from the RPQ, HADS and PCL-C were positively skewed. However, as the data were similarly skewed for each group, the sample sizes were reasonable, the data did not violate the homogeneity of variance assumptions, and ANOVA is considered robust to violations of normality, we considered it appropriate to use parametric analyses. (62, 63)”

43. In the result part absolute numbers e.g. for the RPQ should be interpreted in terms of the given cut-offs in the literature. For example, it should be stated whether patients had a manifest post-concussion syndrome. For definitions see e.g. Riemann et al. 2021 (DOI: 10.3171/2020.9.PEDS20421).

We had stated that 30.5% of the mTBI group endorsed four or more symptoms as being at least a moderate problem as compared to 14.6% of the comparison group but had neglected to explain the meaning of these scores. We have added an explanation of the cut-off scores to the measures section (i.e., a score of 3 or higher on four or more items was used to determine if participants met the DSM-IV symptom criteria for post-concussive disorder). We have used these criteria in previous publications. We have also explained these findings in the results and carried out additional chi-square tests to compare males and females in the mTBI and comparison groups for the RPQ and the HADS. 

44. The current literature should be updated, e.g. include references from doi.org/10.1016/S1474-4422(17)30371-X; doi.org/10.1186/s12955-020-01391-3;
doi.org/10.1007/s11357-020-00273-2

We have carried out another literature search and added more recent references as appropriate (including those suggested by the reviewers and recent publications on sex x age interactions after mTBI from the TRACK-TBI team).

45. I assume that the authors are English native speakers. However, some parts of the manuscript would benefit from scientific/ language editing aiming a concise scientific English language.

We have tried to make the wording more concise throughout.

Reviewer 2

This is an interesting article focusing on the long-term consequences of mild TBI in context of sex or gender. Because of the inclusion of a comparison group, long follow-up, inclusion of mild TBI patients from a broad range of settings, and the analysis of the longitudinal trajectory, this study could contribute to the literature in the field.

However, there are some important limitations and concerns related to the analyses, reporting of the results and interpretation/ implications of the results.

Analysis and reporting of the results:

46. The analyses do not adequately adjust for differences in ethnicity and education, although the authors state themselves in the Introduction that these differences can be related to outcomes.

As the comparison sample was larger than the group with mTBI, we carried out additional matching on occupation. This reduced the sample size to 151 (the same as the mTBI group) and there were no significant differences between the groups for ethnicity or education (or any other demographic variables) as shown in Table 2 (p 18)

47. The characteristics of participants are not reported in sufficient detail: characteristics of mTBI patients (GCS, mechanism of injury, prior TBI etc.) should be added to Table 1., so that it is visible how they relate to attrition.

We have added this information to Table 1 (p16) and compare the sample completing the study at 8 years post-injury with the incidence sample (n=870). There is a large amount of missing data as we were not able to contact all of the incidence sample and the only information available was from medical records.

48. The characteristics of participants are not reported for all subgroups (e.g. group by sex) being compared (Table 2).

We have revised Table 2 to provide information about the demographic characteristics of the sub-groups of participants (p18)

49. Statistically insignificant results and associated average results are not reported, as they should be, at least in Supplement.

We have simplified Table 1 and now only compare the incidence sample with the sample completing the study at 8 years post-injury. The table now includes all of the descriptive and inferential statistics.

The descriptive and inferential statistics for the item-level analysis are included in a supplementary Table (Table S1, p 44-45)

50. It is not clear if other relevant medical history and sociodemographic characteristics were measured in this study: if yes, they should be reported and included in the analyses. If not, that should be discussed in detail as limitation.

Apologies for not including this in the earlier version of the manuscript. We collected a wide range of medical and sociodemographic data (e.g., GCS, CT scan, occupation, main income earner, major medical diagnoses) from the mTBI sample and this has been incorporated into Tables 1 and 2. (p16 and p 18)

51. Sex or gender is very important in this study but it is not explained how exactly was measured.

This is a very good point, sorry for the omission. For cases identified through the hospital or medical provider we extracted demographic details from the medical records and verified when participants were invited to take part in the TBI outcome study. We have added this information to the method section (p9 lines 172-175)

Discussion and conclusion

52. The results are not sufficiently interpreted in regard to study population/ characteristics of the included participants, methodological limitations and the literature in the field. The study limitations should be discussed thoroughly. Plausible explanations and clinical implications of the results should be also discussed.

We have added a strengths and limitations section in the discussion that includes discussion of the study population and how it compares to those identified in the original incidence study. This also includes discussion of our modifications to the RPQ, the suitability of the TBI free comparison group and the use of self-report assessments (p32 lines 549-578).

We have compared the current findings with outcomes at earlier time points, and it seems that symptoms have not improved much from 12 months post-injury. This suggests that early intervention is the key to preventing long-term symptom burden. Of note only 6 participants in our 4 year follow up reported any type of treatment for their injury (beyond the first medical presentation for the injury). (page 30 line 501 onwards). 

Females with repeat mTBI appear most at risk for ongoing symptoms burden (in terms of the proportion exceeding the clinical cut-offs for the RPQ, HADS and PCL-C). The proportion of participants exceeding the clinical cut-offs for the depression/anxiety/PTSD scales far exceeds the number reporting a diagnosed psychiatric condition, suggesting that there may be a significant number of people with undiagnosed mental health issues. Given the complex interplay of psychosocial factors in post-concussion symptoms as well as depression and anxiety, early intervention after concussion to ensure that symptoms have fully resolved may be key to preventing long-term symptom burden. 

In more detail: 

53. Abstract: the conclusion and the implications of your results are not clearly stated.

We have restructured the abstract and added a conclusion and a sentence about the implications of the findings (page 3 lines 46-50):

“Thus, at 8-years post-mTBI, people continued to report high symptom burden, and some impacts on work. Women with mTBI, particularly those with a history of prior TBI had the greatest symptom burden. When treating mTBI it is important to assess TBI history as this may help to identify those at greatest risk of poor long-term outcomes to direct early treatment and intervention.”

54. Introduction: there are couple of other studies in general population, and a recent study in TRACK-TBI data that does include a comparison group and could be mentioned.

Thank you for letting us know about the recent TRACT-TBI publication. We have updated the literature and added several recent publications including those from the CENTER-TBI and TRACK-TBI teams

55. I suggest finding a different wording instead of “mood outcomes”

As suggested by reviewer 1 we have renamed the outcome measures as PROMS and have grouped them under three commonly used categories (as described by Weldring & Smith 2013): symptoms and symptom burden (Rivermead post-concussive symptoms questionnaire), health-related behaviours (Hospital Anxiety and Depression Scale, Posttraumatic Stress Disorder: Civilian Scale) and functional status (Participation with Recombined Tools – Objective, Work Limitations Questionnaire).

56. The authors talk about men and women, but about “sex” and not “gender”. How was this variable measured and how is the term understood by the authors?

Apologies for the lack of precision in the language. We recorded the sex of the TBI cases from medical records, and checked the accuracy of the information with the participants in the outcome studies. We have tried to be more accurate throughout the paper and we are describing biological sex differences, not gender differences.

57. There are differences in education and ethnicity. “The MTBI and Comparison group differed in terms of ethnicity and education, but as these were true group differences it was deemed inappropriate to include these variables as covariates in the analyses.(48, 49)”But it is stated (correctly) in the Introduction that “a range of factors including age, minority ethnicity, lower levels of education, and SESare linked to poorer outcomes” . Therefore, if you would like to compare outcomes in groups with history/no history of TBI, they should be adjusted based on these factors (ethnicity, education), in the way you are incorporating age. Otherwise, you cannot know if the (lack of) difference is due to TBI status or differences in education and ethnicity. In addition, considering the recruitment and data collection process, how can you know that “these were the true group differences”?

We have carried out additional matching with the comparison group and there are now no significant differences between the mTBI and comparison groups. In the previous version of the manuscript we were trying the explain why it wasn’t appropriate to control for the differences between the groups even though they are important in terms of outcomes from TBI. When the covariate and experimental effect are not independent in can produce unreliable analyses, and make the analyses hard to interpret. Essentially, we cannot separate the variances of the effect of mTBI from those of SES or ethnicity. We are aware that a lot of people do use ANCOVA in this way (i.e., adding a covariate to control for group differences), but it is not strictly correct (see Miller & Chapman 2001 for a paper on misuses of ANCOVA). 

58. Descriptive characteristics e.g. sex and ethnicity are not presented for subgroups you analyze (separately for women with/without TBI, for men with/without TBI ). In addition, are there other available information for the groups?

We have included the sociodemographic characteristics of the sub-groups in Table 2 (p 18).

59. What are the characteristics of participants who did and did not complete the 8-year measurement in terms of GCS, mechanism of injury, CT abnormalities?

We have provided additional information in Table 1 (p16). We have simplified the Table and just carried out analyses between the original incidence sample and those completing the assessment at 8-years post-injury. Unfortunately there is quite a lot of missing data as some information was not available in medical records.

60. Reporting results based on p-values is inappropriate. I would like to see average values for all compared subgroups at least in Supplement.

We have included all of the descriptive and inferential statistics in Table 1. Additional information for the item level analyses is reported in the supplementary table (p 44-45)

61. Why the comparison “TBI-free group” was not compared with the 1-month and 1-year results of the mTBI group?

We have removed the longitudinal analyses from the paper, and now include analyses of single and multiple TBI. We could not compare the TBI-free group with the sample at earlier injury points because they would not match on age (the TBI-free group was only recruited at the 8-year time point).

62. - The RPQ was used differently (without the comparison before the injury) than in other TBI studies- that should be noted in the Discussion and related to the observed average scores.

We have noted this as a limitation in the discussion (p33 lines 573-578). Under normal circumstances responses of 1 are not included in the total score, so the scores can be directly compared to other published studies.

63. Is it possible to report and include in the analysis other very important factors: psychiatric and general medical history/ health status? If not, this should be discussed in detail as a limitation. How could that influence the results and comparison between groups?

We have added information about medical diagnoses to Table 2. Unfortunately, we did not collect this information from the comparison group, and have noted this as a limitation in the discussion (page 32 line 566). Interestingly there was no significant differences in the proportion of males and females in the mTBI group in terms of prior diagnoses Table 2 (page 18).

64. The average age is relatively low, and the percentage of prior mTBI and the percentage of mechanical force-induced injuries is high. Does it mean that the study included substantial percentage of sport or work- related injuries in young adults/adolescents? How could this influence the results in the context of age?

The sample was identified as part of an epidemiological study of the incidence and outcome from TBI (across all ages and severities), and therefore reflects the characteristics of the injuries occurring in New Zealand in 2010-2011. As it was an epidemiological study we identified mTBI cases that did not seek medical care, so this may explain why our sample is somewhat different to others that have focused solely on cases presenting to hospital. Rates of TBI were highest in children, adolescents and young adults (0-34 years) (Feigin et al (2013). Of the 1369 cases identified 291 were sustained during sport, and 39% or these were in children aged <16 years and were not included in this study (Theadom et al, 2014). The sample identification process may mean that we have a greater number of ‘mild’ mTBI in our sample as they may not typically be included in other outcomes studies, but then we would expect there to be few long-term effects which doesn’t seem to be the case.

We know that the environment / injury management immediately post-injury is crucial in ensuring a good recovery and to reduce the possibility of subsequent injuries. However, there are numerous papers describing the risk-taking behaviour of adolescents and young adults and this may well be a contributing factor to the relatively poor outcomes observed. In addition, the incidence study was conducted ten years ago. Knowledge about the potential negative consequences of concussion/mTBI has grown enormously over the last ten years (in terms of management and the general public’s understanding of concussion), and it is possible that ten years ago these injuries were not taken as seriously, and things that are now taken for granted (such as time off from sport) did not form part of the recommendations. This has been noted in the discussion (p30, line 514).

65. It is not discussed how the time could influence the results- e.g. maybe sex x group differences exist 3-6 months after injury but disappear with time (maybe not)? 

This is a good point. We have raised this issue in the discussion (p 30, line 507-510). Data from the same cohort at 12 months post-injury showed that female sex was a significant predictor of poor outcomes across a number of domains (RPQ, HRQoL, anxiety and GOS), but this was not the case at 4 years post-injury suggesting that perhaps the sex differences diminish over-time (or it may be due to participant attrition).

66. Moreover, discuss why you would expect symptoms after mild TBI so long as 8 years after injury.

- How the patterns can be explained? Why women with mild TBI showed more PTSD symptoms and time-related work demands? How come that symptoms do not change from 1 year to 8 year?

The revised analysis indicates that a significantly greater proportion of females exceed the cut-offs across a number of measures compared to males. We have suggested that this may be due to either lack of early intervention, or possibly undetected anxiety/depression (p 31 line 536 onwards) which is linked to higher symptom burden. Only 14.9% of females had a formal mental health diagnosis but a much greater proportion exceeded the HADS and PCL cut-offs. The additional analyses revealed that females with multiple TBI appear to be at the greatest risk of poor outcomes. 

67. To what degree could that be related to the characteristics of responders?

We have compared the characteristics of the incidence sample and the those completing the assessment at 8-years post-injury. The 8-year sample appeared to be more highly educated and were more likely to have professional or skilled jobs, but there was a lot of missing data so it is hard to draw firm conclusions. There were no significant differences between the males and females in the mTBI group with regard to other health diagnoses, occupation etc so it seems unlikely that these factors could explain the outcomes. Unfortunately, we did not obtain information about health conditions from the control group, but will in future follow-up studies. It is possible that those who completed the 8 year follow up have more ongoing symptoms that those who didn’t, but this is impossible to assess. In all of our interactions with the participants we stress that we want to know how everyone is doing – whether they think they have ongoing symptoms or not.

68. Age comparisons (age subgroups)- 8 years is a long time. What is more important, the age of injury or 8 years after, considering relevant biological and social factors?

We think age at injury, rather then current age is more important in outcomes. The immediate post-injury environment plays a key role in outcomes from mTBI, particularly not doing too much too soon, and also taking steps to reduce the likelihood of a second injury. In terms of sex differences, the interaction between the chemical cascade produced as a result of injury and sex hormones may also play a role, albeit the research about whether female hormones are protective is contradictory. In the post-injury period, other broader psychosocial factors likely come into play, such as stress, pre-existing health conditions etc.

69. What are the implications of the results for clinical practice and patients with mild TBI?

We think it is crucial to provide appropriate post-injury advice to minimise a subsequent injury, instructions about a gradual return to normal activities, as well as clear information about when (and where) to seek further help and support if the symptoms do not seem to be improving. Females who have prior TBI appear to be a greatest risk of poor outcomes so asking about prior TBI would help identify those at greatest risk of long-term symptoms as well as screening for mental health issues. This information has been incorporated into the discussion (p34, lines 592-596).

We hope these extensive changes have addressed your concerns,

Kind regards,

Professor Nicola Starkey

---

## [Decision Letter · Decision Letter 1]

14 Feb 2022

PONE-D-21-27103R1

Sex differences in outcomes from mild traumatic brain injury eight years post-injury.

PLOS ONE

Dear Dr. Starkey,

Thank you for submitting your manuscript to PLOS ONE. After careful consideration, we feel that it has merit but does not fully meet PLOS ONE’s publication criteria as it currently stands. Therefore, we invite you to submit a revised version of the manuscript that addresses the points raised during the review process.

The following changes are necessary:

Shorten the abstract and specify the conclusions providing the key message of your study.Use the definitions “single mTBI” and “repetitive mTBI” throughout the manuscript.Specify complications that you consider as increased risk in Methods. Include in statistical analysis the method you used to correct for multiple comparisons.Check headers and legends in Tables for clarity and add sample size to Figure Legends.In Discussion elaborate more in detail on the limitations of the study and discuss the impact of gender and age on your findings. Proofread the text for accurate scientific language

We look forward to receiving your revised manuscript.

Kind regards,

Anna-Leena Sirén

Academic Editor

PLOS ONE

Reviewers' comments:

Reviewer's Responses to Questions

**Comments to the Author**

1. If the authors have adequately addressed your comments raised in a previous round of review and you feel that this manuscript is now acceptable for publication, you may indicate that here to bypass the “Comments to the Author” section, enter your conflict of interest statement in the “Confidential to Editor” section, and submit your "Accept" recommendation.

Reviewer #1: All comments have been addressed

Reviewer #2: (No Response)

2. Is the manuscript technically sound, and do the data support the conclusions?

Reviewer #1: Partly

Reviewer #2: Partly

3. Has the statistical analysis been performed appropriately and rigorously? 

Reviewer #1: Yes

Reviewer #2: No

4. Have the authors made all data underlying the findings in their manuscript fully available?

Reviewer #1: No

Reviewer #2: (No Response)

5. Is the manuscript presented in an intelligible fashion and written in standard English?

Reviewer #1: No

Reviewer #2: Yes

6. Review Comments to the Author

Reviewer #1: Dear authors,

thank you very much for your immense work and effort that have improved your manuscript significantly. However, I have minor suggestions that would help to improve the manuscript further:

• Please provide another orthographic check

• Please double check some wording and grammar by another scientific English native proofread

• I suggest adapting the last part of the abstract and the conclusion by being as specific and congruent in terms of the key results and messages.

• I suggest differing between single mTBI and repetitive mTBI throughout the manuscript

Thank you very much for your effort.

Reviewer #2: I congratulate the authors on a thorough revision. The focus of the study is more clear now.

Abstract- It appears to be too extensive, I suggest describing the results in a more condensed/ summarized manner, and indicating only the most important results. Only the p-values have been reported- it would be more appropriate to report less results but more information: F, df, or effect sizes.

Introduction – More relevant studies are included now. However, it would profit from shortening and summarizing.

Methods- Risk of complications after mild TBI: What kind of complications- Intracranial abnormalities or symptoms (based on the factors mentioned it seems intracranial abnormalities)? I suggest describing more clearly.

Multiple comparisons are quite relevant here, particularly because the authors put a lot of emphasis on statistical significance. What did the authors do to to correct for multiple comparisons? It should be described better how the p-values were obtained and reported.

Depression, anxiety and PTSD do not seem part of “health-related behaviors” but rather “symptoms and symptoms burden”.

Results

Table 1- what does “CT scan” mean?

Table 2- is comparison for “main earner” is correct- who is compared? What was tested in “Health”? The sex difference in psychiatric diseases seems considerable! What did you test exactly- it is not clear from the table.

Injury severity is described by sample (incidence/ complete) but not by sex for mTBI. Why?

“In contrast, a significantly greater proportion of women in the mTBI group met the DSM-IV symptom criteria than in the comparison group (X 2 347 (1) = 10.63, p<.01).”- Why in contrast? The difference is in the same direction, and the sample of women is larger.

There is too much emphasis on statistical significance and not on size of differences or effect sizes. The size effects are calculated but not commented, and percentages/ number of participants should be consistently reported.

Figures- sample sizes should be clearly stated.

Discussion- You mention the results of other timepoints of this study, which is appropriate and interesting-they should be introduced more clearly in the Introduction.

“The data was self-report which may be subject to bias”- but these are patient reported outcomes, they should be self-report.

Could the lack of age difference be due to not controlling for injury severity?

Limitations: It would be good to mention different percentages of males and females at 8 years and missing data on injury. In addition, lack of statistical differences between males and females in a relatively small sample does not mean that the differences were not there and do not matter (also, it is not clear if the authors tested differences between group or sex in Table 2)- importantly, there is a considerable difference in the percentage of psychiatric disorders.

7. PLOS authors have the option to publish the peer review history of their article (what does this mean?). If published, this will include your full peer review and any attached files.

Reviewer #1: **Yes: **Katrin Rauen

Reviewer #2: No

---

## [Author Response · Author response to Decision Letter 1]

31 Mar 2022

31st March 2022

Dear Professor Sirén,

Re: PONE-D-21-27103R1

Sex differences in outcomes from mild traumatic brain injury eight years post-injury.

Thank you for the comments on the revised version of the paper. We have outlined how we have addressed your comments and those of the reviewers in the letter below. Please note that the page and line numbers refer to the tracked changes version of the paper. Please let me know if you need any further information,

Kind regards,

Professor Nicola Starkey 

Responses to the Editor’s comments

1. Shorten the abstract and specify the conclusions providing the key message of your study.

We have shortened the abstract and focused on reporting the most important results. We have incorporated the statistical results into the abstract as appropriate (as requested by R2).

2. Use the definitions “single mTBI” and “repetitive mTBI” throughout the manuscript.

We have changed this throughout the manuscript

3. Specify complications that you consider as increased risk in Methods. 

The sub-classification of mTBI severity is based on the risk of developing intracranial hematoma. This has been included in the methods (p9, line 172).

4. Include in statistical analysis the method you used to correct for multiple comparisons.

We have explained that we used a Bonferronni correction (p15 line 314) and have added/commented on effect size throughout the results

5. Check headers and legends in Tables for clarity and add sample size to Figure Legends.

We have provided more detailed descriptions for the table headers, and added more detail to the Figure legends, including the sample size and the cut-off scores. We have also indicated on the Figures which groups are significantly different.

6. In Discussion elaborate more in detail on the limitations of the study and discuss the impact of gender and age on your findings. 

We have elaborated on the limitations of the study, in particular the lower percentage of males, and those of Maori ethnicity in the 8 year follow up sample. We have also highlighted that our sample size limited our ability to explore fully the effects of age on outcomes (page 37).

7. Proofread the text for accurate scientific language

We have carefully proof read the paper.

Responses to Reviewer #1: 

1. Please provide another orthographic check

We have double checked the spelling and abbreviations

2. Please double check some wording and grammar by another scientific English native proofread

The wording has been simplified throughout

3. I suggest adapting the last part of the abstract and the conclusion by being as specific and congruent in terms of the key results and messages.

The abstract has been shortened to make it more focused on the key findings and conclusions (pages 2-3).

4. I suggest differing between single mTBI and repetitive mTBI throughout the manuscript

Thank you for that suggestion – it’s a much simpler way to refer to the two groups. We have changed this throughout the manuscript 

Response to Reviewer #2: 

I congratulate the authors on a thorough revision. The focus of the study is more clear now.

1. Abstract- It appears to be too extensive, I suggest describing the results in a more condensed/ summarized manner, and indicating only the most important results. Only the p-values have been reported- it would be more appropriate to report less results but more information: F, df, or effect sizes.

We have shortened the abstract as suggested and focus on the key findings. We have incorporated the statistical results as appropriate.

2. Introduction – More relevant studies are included now. However, it would profit from shortening and summarizing.

We have shortened the introduction and removed the less relevant information (e.g. p 5 lines 87-92; p 6 lines 115-123)

3. Methods- Risk of complications after mild TBI: What kind of complications- Intracranial abnormalities or symptoms (based on the factors mentioned it seems intracranial abnormalities)? I suggest describing more clearly.

Sorry for the lack of clarity. The categorisation of mTBI was based on the risk of developing intracranial hematoma. This has been clarified p 9 line 172

4. Multiple comparisons are quite relevant here, particularly because the authors put a lot of emphasis on statistical significance. What did the authors do to correct for multiple comparisons? It should be described better how the p-values were obtained and reported.

The Bonferonni correction was used to adjust for multiple comparisons for the item level analyses (i.e., the p value, .05, was divided by the number of items on the measure). This has been clarified p 15 line 314

5. Depression, anxiety and PTSD do not seem part of “health-related behaviors” but rather “symptoms and symptoms burden”.

We have removed the title “health-related behaviours” so that depression, anxiety and PTSD are part of the “symptoms and symptom burden” section

Results

6. Table 1- what does “CT scan” mean?

It is the number (%) of cases that had a CT scan. We have changed the row heading to ‘CT scan conducted’ to make it clear to the reader 

7. Table 2- is comparison for “main earner” is correct- who is compared? 

The analysis compares males and females who are the main income earners in the TBI and comparison group. A contingency table was produced to calculate the chi square statistic and z tests were calculated to compare column proportions (i.e., between males and females). We have removed the ‘No’ row for clarity.

8. What was tested in “Health”? The sex difference in psychiatric diseases seems considerable! What did you test exactly- it is not clear from the table.

The health section of Table 2 provides details of the participants previous or co-morbid diagnoses. Participants were assigned a code based on the health conditions they reported (e.g. cardiovascular disease, diabetes, seizure). A contingency table was produced for the observed frequencies for each diagnosis for males and females, and a z-test was used to compare cell counts across the column proportions (i.e. between males and females). We have now re-run the analyses as separate tests for each diagnosis for clarity and added Cramer’s V as a measure of effect size. The difference in psychiatric diagnosis was not statistically significant but did show a small to moderate effect size.

9. Injury severity is described by sample (incidence/ complete) but not by sex for mTBI. Why?

This was an oversight and has now been added to Table 2. There was no significant association between injury severity and sex in the mTBI group.

10. “In contrast, a significantly greater proportion of women in the mTBI group met the DSM-IV symptom criteria than in the comparison group (X 2 347 (1) = 10.63, p<.01).”- Why in contrast? The difference is in the same direction, and the sample of women is larger.

We have reworded this section of the results and simplified the wording throughout the results section to avoid contradictory statements. 

11. There is too much emphasis on statistical significance and not on size of differences or effect sizes. The size effects are calculated but not commented, and percentages/ number of participants should be consistently reported. Figures- sample sizes should be clearly stated

We have included comment on the effect sizes throughout the results section and calculated Cramer’s V as an effect size for the Chi-square tests. In most cases the statistically significant findings had larger effect sizes. 

We have reported the number and percentage of participants in the text only where it is not reported in the table. We have added the total sample size to Tables 3 and 4. Both Tables includes a footnote to indicate that the sample size for the work limitations questionnaire was smaller as the measure was only completed by those in employment. We have also expanded the descriptions of the Tables and added more details to the Figure legends, including the sample sizes.

12. Discussion- You mention the results of other timepoints of this study, which is appropriate and interesting-they should be introduced more clearly in the Introduction.

We have amended the introduction to highlight earlier findings from the same cohort (by stating in our previous research, or findings from the BIONIC cohort) – p4, line 78; p 5 line 85, line 99

13. “The data was self-report which may be subject to bias”- but these are patient reported outcomes, they should be self-report.

We agree and have deleted this statement. It was included at the suggestion of a previous reviewer.

14. Could the lack of age difference be due to not controlling for injury severity?

We think it is unlikely – we ran a one-way Anova comparing the mean age of each injury severity group and it was not significant and had a small effect size. I do think sample size is an issue though – we need a much larger sample to properly explore the impact of age at injury, and I have added a comment about this in the limitations section (page 37, line 651-655)

15. Limitations: It would be good to mention different percentages of males and females at 8 years and missing data on injury.

We have added further information about the differences between the incidence cohort and the 8-year sample, as well as the lack of information about injury severity (i.e. GCS) in the discussion section (p36-37, line 636-641, line 644).

16. In addition, lack of statistical differences between males and females in a relatively small sample does not mean that the differences were not there and do not matter (also, it is not clear if the authors tested differences between group or sex in Table 2)- importantly, there is a considerable difference in the percentage of psychiatric disorders.

 We agree and have further emphasised our small sample as a limitation of the study (p 37 line 651-655. For Table 2 we carried out chi-square tests and used z test to compare the column proportions (i.e., sex differences). There were no statistically significant differences between the proportions of males and females for either group.

---

## [Decision Letter · Decision Letter 2]

4 May 2022

PONE-D-21-27103R2Sex differences in outcomes from mild traumatic brain injury eight years post-injury.PLOS ONE

Dear Dr. Starkey,

Thank you for submitting your manuscript to PLOS ONE. After careful consideration, we feel that it has merit but does not fully meet PLOS ONE’s publication criteria as it currently stands. Therefore, we invite you to submit a revised version of the manuscript that addresses the points raised during the review process.

Please clarify the unclear sentence “38.1% of the sample met the cut-offs for anxiety and/or depression on the HADS at 12 months post-injury (similar to rates observed in community samples in our previous research; 7.7% for depression, 13.6% for anxiety which decreased to 32.1% four 72 years post-injury” in the Introduction as suggested by the Reviewer 1.

Please in methods that mTBI was not defined strictly according to the American Congress of Rehabilitation Medicine criteria and cite the correct literature for these criteria.

Please use systematically the term mTBI for mild traumatic brain injury.

Please correct the orthographic mistakes throughout the manuscript.

We look forward to receiving your revised manuscript.

Kind regards,

Anna-Leena Sirén

Academic Editor

PLOS ONE

Journal Requirements:

Reviewers' comments:

Reviewer's Responses to Questions

**Comments to the Author**

1. If the authors have adequately addressed your comments raised in a previous round of review and you feel that this manuscript is now acceptable for publication, you may indicate that here to bypass the “Comments to the Author” section, enter your conflict of interest statement in the “Confidential to Editor” section, and submit your "Accept" recommendation.

Reviewer #1: All comments have been addressed

2. Is the manuscript technically sound, and do the data support the conclusions?

Reviewer #1: Yes

3. Has the statistical analysis been performed appropriately and rigorously? 

Reviewer #1: Yes

4. Have the authors made all data underlying the findings in their manuscript fully available?

Reviewer #1: Yes

5. Is the manuscript presented in an intelligible fashion and written in standard English?

Reviewer #1: Yes

6. Review Comments to the Author

Reviewer #1: Dear authors,

thank you very much for your efforts that clarified your manuscript. However, I have still minor suggestions that would help to improve the manuscript further:

• There are still minor orthographic mistakes, such as spaces, double points, etc. The authors should thoroughly double check this or it should be done by PLOS One during the final editing. Please see further minor points that need corrections:

Within the introduction (page 4, line 67) the term mild TBI should be replaced by mTBI. The sentence (page 5, line 73) should start with a capital letter, thus MTBI. NZ population (page 5, line 80) should be introduced as New Zealand… when used for the first time.

• Content-wise: the intro (p.4-5, lines 67-72) should be clarified as the following sentence is somehow confusing: “38.1% of the sample met the cut-offs for anxiety and/or depression on the HADS at 12 months post-injury (similar to rates observed in community samples in our previous research; 7.7% for depression, 13.6% for anxiety which decreased to 32.1% four 72 years post-injury”

Most probably the authors meant that the initial 38.1% of mixed anxiety and/or depressive symptoms decreased by 6% between year 1 and 4 after mTBI. Why did the authors mention the community samples, is this sample a mixed sample of TBI positive and negative humans? It would be particularly of interest whether the portion of psychiatric symptoms was increased in comparison to the general population. Please, could the authors clarify this point?

• Methods (page 8, lines 141-143): as mTBI was not clearly defined according to the ACRM criteria, this should be stated in the limitation part (lack of information on the time of loss of consciousness). I am very sorry for not having this mentioned before. Please see references: Medicine ACoR. Definition of mild traumatic brain injury. Journal of Head Trauma Rehabilitation. 1993;8(3):86-87; Silverberg ND, Iverson GL. Expert Panel Survey to Update the American Congress of Rehabilitation Medicine Definition of Mild Traumatic Brain Injury. Archives of physical medicine and rehabilitation. 2021;102(1):76-86;

Besides of the definition of mild TBI, my comments are minor issues. Thanks for your effort researching long-term outcome after mTBI that provides important information for patients, clinicians, and future research.

7. PLOS authors have the option to publish the peer review history of their article (what does this mean?). If published, this will include your full peer review and any attached files.

Reviewer #1: **Yes: **Katrin Rauen, MD, FEBN

---

## [Author Response · Author response to Decision Letter 2]

9 May 2022

Dear Anna-Leena Sirén,

Re:PONE-D-21-27103R2

Sex differences in outcomes from mild traumatic brain injury eight years post-injury.

PLOS ONE

Thank you for the helpful editor’s and reviewer’s comments on our revised manuscript. We have explained how we have addressed each of the comments below and hope the manuscript is now acceptable for publication in PlOS ONE.

Editor’s comments

1. Please clarify the unclear sentence “38.1% of the sample met the cut-offs for anxiety and/or depression on the HADS at 12 months post-injury (similar to rates observed in community samples in our previous research; 7.7% for depression, 13.6% for anxiety which decreased to 32.1% four 72 years post-injury” in the Introduction as suggested by the Reviewer 1.

We have revised the wording as follows:

“38.1% of the sample met the cut-offs for anxiety and or depression on the HADS at 12 months post-injury (2.2% depression, 25.5% anxiety, 10.2% depression and anxiety) which decreased to 32.1% four years post-injury (7.7% depression only, 13.6 % anxiety only and 11.8% depression and anxiety), similar to rates observed in community samples.”

2. Please in methods that mTBI was not defined strictly according to the American Congress of Rehabilitation Medicine criteria and cite the correct literature for these criteria.

We have added this to the limitations sections as suggested by the reviewer (p 36, line 598-600):

“contact. In addition, we used WHO criteria to define mTBI, not the American Congress of Rehabilitation, which includes the presence of observable signs and subjective symptoms (dizziness, confusion) [60, 61]. “

3. Please use systematically the term mTBI for mild traumatic brain injury.

Corrected throughout

4. Please correct the orthographic mistakes throughout the manuscript.

Corrected

 

Response to the reviewer

1. There are still minor orthographic mistakes, such as spaces, double points, etc. The authors should thoroughly double check this or it should be done by PLOS One during the final editing. 

We have checked this again and corrected all of the issues we could find including the reference format and punctuation around the brackets.

2. Within the introduction (page 4, line 67) the term mild TBI should be replaced by mTBI. 

 We have corrected this

3. The sentence (page 5, line 73) should start with a capital letter, thus MTBI. 

 Corrected

4. NZ population (page 5, line 80) should be introduced as New Zealand… when used for the first time.

 Corrected

5. Content-wise: the intro (p.4-5, lines 67-72) should be clarified as the following sentence is somehow confusing: “Most probably the authors meant that the initial 38.1% of mixed anxiety and/or depressive symptoms decreased by 6% between year 1 and 4 after mTBI. Why did the authors mention the community samples, is this sample a mixed sample of TBI positive and negative humans? It would be particularly of interest whether the portion of psychiatric symptoms was increased in comparison to the general population. Please, could the authors clarify this point?

Apologies for this – we agree it doesn’t make sense and I suspect it happened when we shortened the introduction. Your interpretation is correct, rates decreased by 6% between 1- and 4-years post-injury. It was not a mixed TBI and community sample. At 4 years post-injury the rates of anxiety and depression were similar to those observed in community samples. 

We have now reworded it as follows and added details about rates for anxiety, depression, and comorbid illness separately (pages 4-5 lines 71- 76):

“38.1% of the sample met the cut-offs for anxiety and or depression on the HADS at 12 months post-injury (2.2% depression, 25.5% anxiety, 10.2% depression and anxiety) which decreased to 32.1% four years post-injury (7.7% depression only, 13.6 % anxiety only and 11.8% depression and anxiety), similar to rates observed in community samples.”

6. Methods (page 8, lines 141-143): as mTBI was not clearly defined according to the ACRM criteria, this should be stated in the limitation part (lack of information on the time of loss of consciousness). I am very sorry for not having this mentioned before. Please see references: Medicine ACoR. Definition of mild traumatic brain injury. Journal of Head Trauma Rehabilitation. 1993;8(3):86-87; Silverberg ND, Iverson GL. Expert Panel Survey to Update the American Congress of Rehabilitation Medicine Definition of Mild Traumatic Brain Injury. Archives of physical medicine and rehabilitation. 2021;102(1):76-86;

Thank you for the citations. We have added the following to the limitations section (p 36, line 598-600):

“contact. In addition, we used WHO criteria to define mTBI, not the American Congress of Rehabilitation, which includes the presence of observable signs and subjective symptoms (dizziness, confusion) [60, 61]. “

Please let me know if you need any additional information,

Yours sincerely,

Professor Nicola Starkey 

Email: nstarkey@waikato.ac.nz

---

## [Decision Letter · Decision Letter 3]

16 May 2022

Sex differences in outcomes from mild traumatic brain injury eight years post-injury.

PONE-D-21-27103R3

Dear Dr. Starkey,

We’re pleased to inform you that your manuscript has been judged scientifically suitable for publication and will be formally accepted for publication once it meets all outstanding technical requirements.

Kind regards,

Anna-Leena Sirén

Academic Editor

PLOS ONE

Additional Editor Comments (optional):

Reviewers' comments:

Reviewer's Responses to Questions

**Comments to the Author**

1. If the authors have adequately addressed your comments raised in a previous round of review and you feel that this manuscript is now acceptable for publication, you may indicate that here to bypass the “Comments to the Author” section, enter your conflict of interest statement in the “Confidential to Editor” section, and submit your "Accept" recommendation.

Reviewer #1: All comments have been addressed

2. Is the manuscript technically sound, and do the data support the conclusions?

Reviewer #1: Yes

3. Has the statistical analysis been performed appropriately and rigorously? 

Reviewer #1: Yes

4. Have the authors made all data underlying the findings in their manuscript fully available?

Reviewer #1: Yes

5. Is the manuscript presented in an intelligible fashion and written in standard English?

Reviewer #1: Yes

6. Review Comments to the Author

Reviewer #1: Dear Prof. Starkey,

thank you very much for your thorough work and important contribution in terms of return to work after mild TBI.

7. PLOS authors have the option to publish the peer review history of their article (what does this mean?). If published, this will include your full peer review and any attached files.

Reviewer #1: **Yes: **Katrin Rauen, MD, FEBN, Consultant Neurology, Psychiatry & Psychotherapy

---

## [Editor Report · Acceptance letter]

18 May 2022

PONE-D-21-27103R3 

Sex differences in outcomes from mild traumatic brain injury eight years post-injury. 

Dear Dr. Starkey:

I'm pleased to inform you that your manuscript has been deemed suitable for publication in PLOS ONE. Congratulations! Your manuscript is now with our production department. 

Kind regards, 

on behalf of

Dr. Anna-Leena Sirén 

Academic Editor

PLOS ONE